# Cryptic evolved melts beneath monotonous basaltic shield volcanoes in the Galápagos Archipelago

Michael J. Stock [1,2✉], Dennis Geist[3,4], David A. Neave [5], Matthew L. M. Gleeson [2], Benjamin Bernard[6], Keith A. Howard [7], Iris Buisman[2] & John Maclennan[2]

Many volcanoes erupt compositionally homogeneous magmas over timescales ranging from decades to millennia. This monotonous activity is thought to reflect a high degree of chemical homogeneity in their magmatic systems, leading to predictable eruptive behaviour. We combine petrological analyses of erupted crystals with new thermodynamic models to characterise the diversity of melts in magmatic systems beneath monotonous shield volcanoes in the Galápagos Archipelago (Wolf and Fernandina). In contrast with the uniform basaltic magmas erupted at the surface over long timescales, we find that the sub-volcanic systems contain extreme heterogeneity, with melts extending to rhyolitic compositions. Evolved melts are in low abundance and large volumes of basalt flushing through the crust from depth overprint their chemical signatures. This process will only maintain monotonous activity while the volume of melt entering the crust is high, raising the possibility of transitions to more silicic activity given a decrease in the crustal melt flux.

[1] Department of Geology, Trinity College Dublin, Dublin, Ireland. [2] Department of Earth Sciences, University of Cambridge, Cambridge, UK. [3] Department of Geology, Colgate University, Hamilton, NY, USA. [4] Division of Earth Sciences, U.S. National Science Foundation, Alexandria, VA, USA. [5] Department of Earth and Environmental Sciences, The University of Manchester, Manchester, UK. [6] Instituto Geofísico, Escuela Politécnica Nacional, Quito, Ecuador. [7] United States Geological Survey, Menlo Park, CA, USA. ✉email: Michael.Stock@tcd.ie

Volcanoes are underlain by complex and dynamic magmatic systems that often span tens of kilometres of crust from the Moho to the near-surface[1,2]. As melts ascend through the lithosphere they undergo diverse processes, including crystallisation, volatile exsolution, assimilation of the surrounding country rock, mixing between different magma batches, and interaction with mush crystallised from previous magmas[3–5]. These processes modify magma compositions, creating significant diversity in the chemistry of igneous rocks observed at the Earth's surface[6]. Despite this multitude of sub-surface processes, many magmatic systems exhibit remarkably monotonous volcanic behaviour, erupting chemically homogeneous liquids over long timescales (several decades to millennia)[7–14]. The causes of monotonous volcanism are not currently well understood, but constraining the architecture and dynamics of monotonous systems is essential for determining their longevity and identifying the potential for future changes in eruptive behaviour that may result in more hazardous activity[15].

Current models for producing chemically monotonous eruption sequences typically involve: (1) uniformity by reactive filtration[8], whereby distinct magma batches interact with surrounding gabbroic material to form chemically homogeneous products or (2) uniformity by processing, whereby successive batches of ascending magma evolve under the same $P$-$T$ conditions[16]. The first model typically requires hot primitive melts to cool and react as they ascend through a super-solidus mush column, buffering their temperature and composition[1,8,17,18]. In many cases, however, the products of monotonous eruption series contain multiple macrocryst populations that record complex growth histories. Geophysical and geobarometric constraints on the crustal structure beneath monotonous volcanoes are also inconsistent with single large storage regions[9,19–22]. The second model often includes repeated recharge of upper crustal magma bodies by chemically consistent melts ascending from depth, maintaining the crustal system within a narrow temperature and compositional range[7,10,11,23]. In this case, monotonous activity

necessitates an exact thermal balance between the heat supplied by ascending magma and the heat lost by advection and eruption.

The western Galápagos Archipelago is an ideal location for studying compositionally monotonous volcanism because it hosts several volcanoes that have erupted near-homogeneous basaltic magmas for several millennia[8,17,24,25]. These include Fernandina volcano (on the island of the same name) and Wolf, Darwin, Ecuador, and Sierra Negra volcanoes on Isabela Island (Fig. 1a)[8,26]. All of the basalts emplaced during monotonous Galápagos eruptions have undergone extensive olivine, clinopyroxene and plagioclase crystallisation[8,26,27]. However, historic lavas at each individual centre have pre-eruptive storage temperatures within a range of only ~22–30 °C[17,28]. Monotonous volcanoes on Fernandina and Isabela are located near the centre of the inferred Galápagos plume[29]. The prevailing hypothesis is that the uniformity in their erupted products is related to the flux of mantle-derived magma entering the crust; the volcanoes receive enough thermal input to sustain thick, thermochemically steady-state gabbroic mush zones in the mid-crust, which interact with ascending magmas, buffering their temperature and composition[8]. More compositionally diverse Galápagos volcanoes are thought to lack such large mush zones: Cerro Azul is also proximal to the inferred plume centre but erupts basalts with a greater range of MgO concentrations, likely because there has been insufficient magma input to the crust for a thermally stable magmatic system to develop[30]; Alcedo is downstream from the plume centre and has produced dacitic and rhyolitic eruptions, which may reflect the mush reaching a phase where melts entering the crust are able to cool and fractionate[31]. Harpp and Geist[26] extend this geographic trend to volcanoes in the eastern archipelago, suggesting that their more primitive and compositionally variable eruptions reflect the absence of sustained magma plumbing systems and thus limited crustal processing.

Recent Galápagos eruptions have afforded geophysical constraints on magma storage depths[19,21,32,33], making them good targets for understanding the processes responsible for

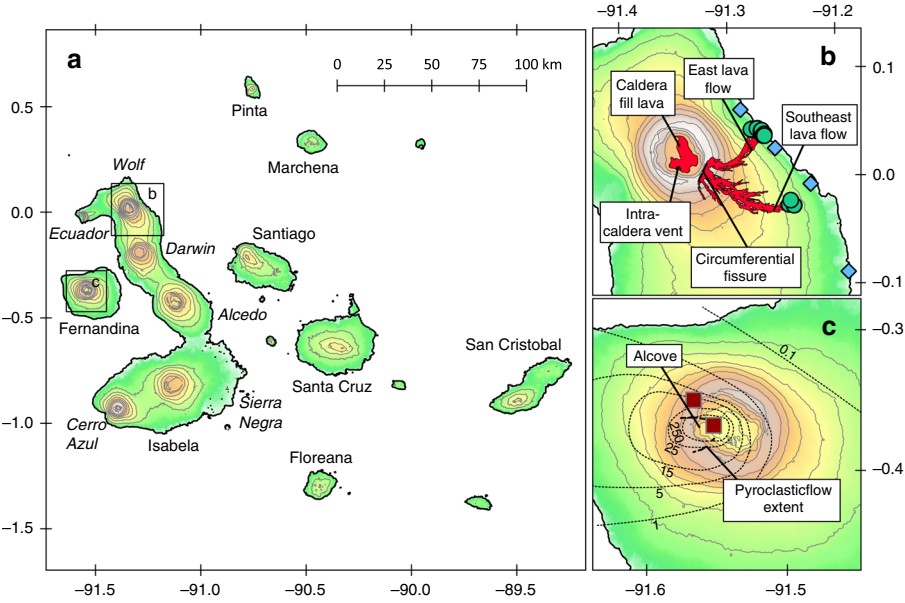

**Fig. 1 Locations of Wolf and Fernandina volcanoes and the sampling sites in this study. a** Regional map of the Galápagos Archipelago showing the different volcanic centres on Isabela Island. Boxes show the locations of maps (**b**) and (**c**). **b** Detailed map of Wolf volcano showing the locations of the 2015 circumferential fissure and intra-caldera vent. The 2015 lava flow extent is in red, after Bernard et al.[71]. The sampling locations of the lavas and tephra analysed in this study are shown as green circles and blue diamonds, respectively. **c** Detailed map of Fernandina showing the extent of the pyroclastic flow (dashed line) and isopachs of the total tephra (in cm; dotted lines) produced during the 1968 eruption, after Howard et al.[48]. The sampling locations of the nodules analysed in this study are shown as dark red squares. Contours are 200 m.

homogenising erupted liquids. In this study, we use detailed microanalyses of mineral phases to constrain magmatic processes at Wolf and Fernandina volcanoes (Fig. 1), which have erupted monotonous basaltic melts for several millennia (Fig. 2)[8,17,24,25]. We focus on integrating petrographic and geochemical observations of pre-eruptive compositional heterogeneity with thermodynamic models that reveal the range of magmas present in the sub-volcanic systems. This allows us to identify the sub-volcanic process that regulates the diversity of erupted liquids. By comparing Wolf and Fernandina with other Galápagos volcanoes, we identify controls on the transition between monotonous and more compositionally varied volcanism.

## Results and discussion

**Samples and petrography.** The samples used in this study include basaltic lava and reticulitic tephra from the 2015 Wolf eruption and gabbroic nodules from the 1968 Fernandina eruption (see Supplementary Note 1 for eruption chronologies). They were selected because they have existing petrological constraints that provide information on their storage depths within their respective sub-volcanic systems. The Wolf lavas are from the circumferential fissure phase of the eruption and contain ≳50 μm, euhedral–subhedral plagioclase (~5 vol.%), clinopyroxene (~2 vol.%) and olivine (<1 vol.%) macrocrysts within a microcrystalline groundmass. The macrocrysts occur in three distinct textural associations: (1) isolated phenocrysts surrounded by groundmass; (2) glomerocrystic aggregates containing plagioclase + clinopyroxene ± olivine (Fig. 3a, b); (3)

plagioclase aggregates attached by synneusis that share common rims. The Wolf tephra was produced during an initial explosion and contains the same macrocryst assemblage as the lava samples, except that anhedral quartz is also present in very low abundance (≪1 vol.%; Fig. 3c) and clinopyroxene crystals occasionally contain ilmenite inclusions (Fig. 3c–e). The Fernandina nodules were exhumed during a hydromagmatic paroxysm and have a phaneritic texture, indicating slow cooling, with a gabbroic mineral assemblage comprising euhedral plagioclase (60–70 vol.%), subhedral clinopyroxene (≤20 vol.%) and subhedral–anhedral olivine (2–15 vol.%). Plagioclase occurs both as independent grains and as inclusions within earlier-crystallising phases (usually clinopyroxene). Miarolitic cavities (<2 mm diameter) are also present, with clinopyroxene lining many of the void walls. Hydrothermally altered samples exhibit secondary pyrite, iddingsite replacement of olivine and epidotisation of other primary magmatic minerals (Fig. 3f–g).

**Homogeneous erupted liquids.** We compiled published whole-rock and matrix glass analyses from historic Wolf and Fernandina eruptions to evaluate the degree of variability in erupted liquid compositions (Fig. 2; see Supplementary Note 2 for data sources). The $Mg\#_{liq}$ ($Mg\#_{liq}$ = atomic $Mg/[Mg + Fe^{2+}]\cdot100$; see Methods) interquartile ranges of subaerial lavas from Wolf and Fernandina are only 4.85 (median = 52.4) and 2.86 (median = 53.1), respectively. Some lavas from both volcanoes contain accumulated plagioclase, which slightly decreases their whole-rock $Mg\#_{liq}$, and a small number of lavas from Fernandina have

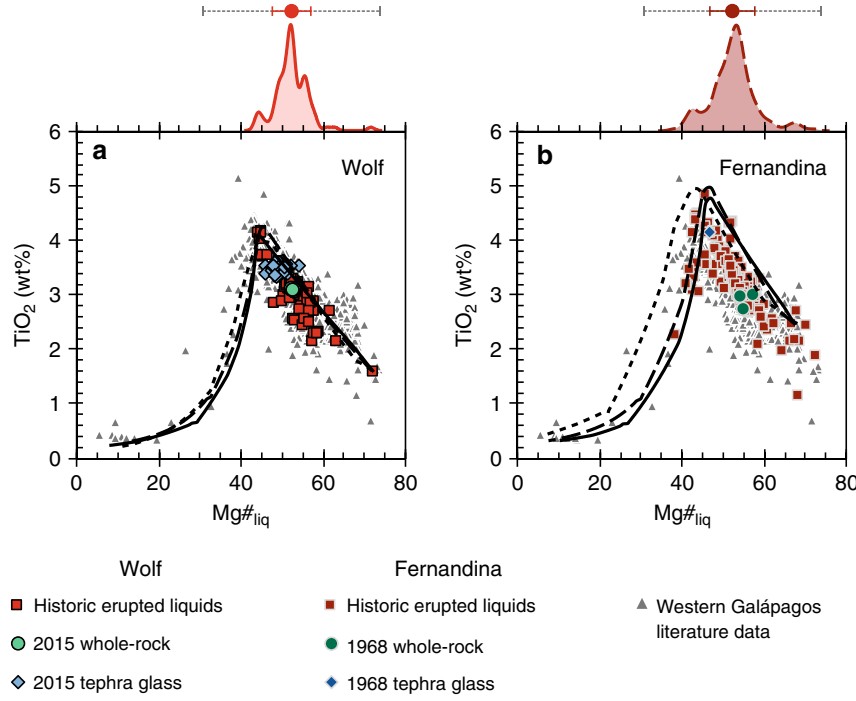

**Fig. 2 Erupted liquid compositions from the Galápagos Archipelago.** $TiO_2$ vs. $Mg\#_{liq}$ of erupted liquids at **a** Wolf volcano and **b** Fernandina. The points show: whole-rock, tephra glass, submarine glass and melt inclusion literature data from all volcanoes in the western Galápagos Archipelago (excluding intrusive rocks and plagioclase-ultraphyric lavas); historic erupted liquids (i.e. whole-rocks, tephra glasses and submarine glasses) from Wolf and Fernandina; and whole-rock and tephra glass data from the 2015 Wolf and 1968 Fernandina eruptions (see legend). References for Wolf and Fernandina historic erupted liquids and literature data from all western Galápagos volcanoes are in Supplementary Note 2. Glass data for the 2015 Wolf eruption are from Stock et al.[19] and whole-rock data are from this study. Glass and whole-rock data for the 1968 Fernandina eruption are from Allan and Simkin[28]. Characteristic 2σ analytical uncertainties for our whole-rock analyses are less than the size of a data point. The black lines show liquid lines of descent calculated using Rhyolite-MELTS at 50 MPa (solid line), 300 MPa (dashed line) and 500 MPa (dotted line). The kernel density estimates above each panel show the $Mg\#_{liq}$ distribution of the historic erupted liquids from Wolf and Fernandina, the red points show their median $Mg\#_{liq}$, and the red bars show their $Mg\#_{liq}$ interquartile range. The grey dotted bars above each panel show the $Mg\#_{liq}$ interquartile range of historic lavas from Santiago[26] (centred around the Wolf and Fernandina medians) for comparison.

## Wolf lava — lower crust

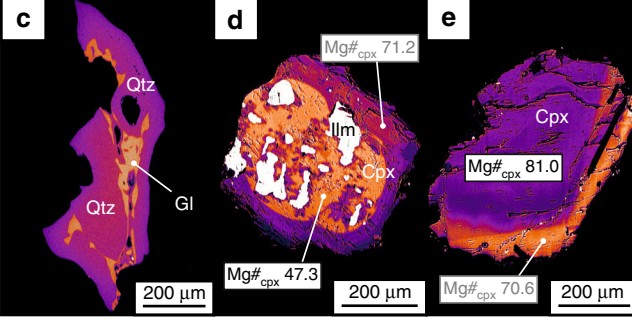

## Wolf tephra — upper crust (?)

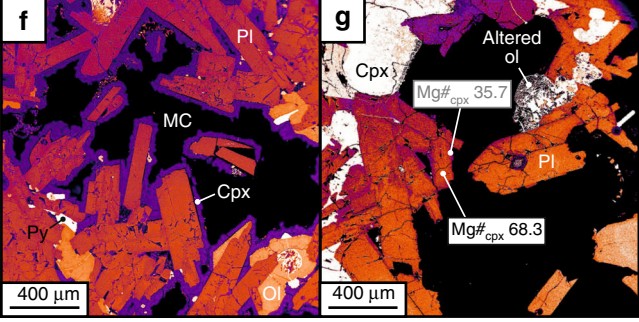

## Fernandina nodule — upper crust

**Fig. 3 Petrographic evidence for heterogeneous liquids and multi-level magma storage. a** Quantitative Evaluation of Minerals by SCANing electron microscopy (QEMSCAN) phase map and **b** Ca map of a glomerocryst in a 2015 Wolf lava sample. The values in (**b**) show the An# of plagioclase zones. **c–e** False colour backscattered electron (BSE) images of quartz and clinopyroxene crystals in tephra samples from the 2015 Wolf eruption (**d** and **e** are adapted from Stock et al.[19]). The values in (**d**) and (**e**) show the Mg#$_{cpx}$ of clinopyroxene zones. **f**, **g** False colour BSE images of nodule samples from the 1968 Fernandina eruption, showing the phaneritic texture, a miarolitic cavity and evidence of hydrothermal alteration. An# and Mg#$_{cpx}$ values in black and grey show averages from crystal cores and rims, respectively. Ol - olivine, Cpx - clinopyroxene, Pl - plagioclase, Qtz - quartz, Gl - glass, Ilm - ilmenite, Py - pyrite, MC - miarolitic cavity.

accumulated olivine, which significantly increases their bulk Mg#$_{liq}$[17,28,34]. Additionally, a few spatially restricted submarine lavas erupted on the southwest flank of Fernandina have more evolved (low MgO, high $K_2O$; evolved series) compositions[34]; including submarine samples increases the Fernandina Mg#$_{liq}$ interquartile range to 5.56 (median = 52.8). Nevertheless, the Mg#$_{liq}$ interquartile ranges of historic eruptions at Wolf and Fernandina are extremely limited and reflect a remarkable degree of homogeneity (cf. Mg#$_{liq}$ interquartile ranges of 12.7 at Santa Cruz and 21.4 at Santiago in the eastern Galápagos

Archipelago[8,26]; Fig. 2), especially given that the lavas represent thousands of years of eruptive activity[17,25].

We analysed 18 whole-rock samples from the 2015 Wolf eruption (Fig. 2; ref. [19]; Supplementary Dataset 1). These do not show any significant compositional variability outside of uncertainty, despite sampling lava flows that erupted at different times through the circumferential fissure phase of the eruption. The bulk samples have an average Mg#$_{liq}$ of 53.0 ± 0.23 (1σ of all whole-rock analyses), which is higher than most previous measurements of the 2015 tephra glass (Mg#$_{liq}$ = 45.4 ± 1.22 [1σ of all glass analyses]; Fig. 2a)[19]. Although there are fewer data for the 1968 Fernandina eruption, the scoria glass also has a lower Mg#$_{liq}$ than the bulk lava (Fig. 2b)[28]. These more evolved glass compositions likely reflect minor pre-eruptive crystallisation of the bulk magmas. Whole-rock analyses from the 2015 Wolf eruption have similar $Al_2O_3$ contents to the tephra glass and most historic erupted liquids, indicating that there was no significant plagioclase accumulation (Supplementary Fig. 1). In contrast, bulk lavas from the 1968 Fernandina eruption do have elevated $Al_2O_3$ concentrations, demonstrating that they have accumulated plagioclase (Supplementary Fig. 2). Both the matrix glass and whole-rock Mg#$_{liq}$ of material erupted in 2015 and 1968 fall within the interquartile ranges of liquids historically erupted from Wolf and Fernandina (Fig. 2).

**Heterogeneous mineral compositions**. We collected ~800 plagioclase and ~90 olivine analyses from the 2015 Wolf eruption (Fig. 4; Supplementary Datasets 2, 3), which we have integrated with ~500 clinopyroxene analyses from the same samples reported by Stock et al.[19]. We also collected ~160 plagioclase, ~100 clinopyroxene and ~80 olivine analyses from 1968 Fernandina nodules (Fig. 4; Supplementary Datasets 4–6) and compare these with mineral analyses from historic Fernandina lavas[28,35]. In contrast with the monotony of erupted liquid compositions, these mineral analyses show striking compositional diversity.

In the Wolf samples, plagioclase An# (An# = atomic Ca/[Na + Ca + K]·100) ranges from 48.8 to 86.8. Tephra crystals and lava glomerocrysts extend to the lowest values (Fig. 4a). Regardless of textural association, plagioclase An# kernel density estimates (KDEs) are asymmetric, with the highest peaks at high anorthite contents (An# = 78–82) and long tails extending to lower An#. The Mg#$_{cpx}$ (Mg#$_{cpx}$ and Mg#$_{ol}$ = atomic Mg/[Mg + Fe*]·100, where Fe* is $Fe^{2+}$ + $Fe^{3+}$; see "Methods") of phenocrystic and glomerocrystic clinopyroxene ranges from 71.8 to 84.6. The largest Mg#$_{cpx}$ KDE peak for the clinopyroxene phenocrysts is at 74.5, with a subsidiary peak at 82.3, whereas the largest peak for the glomerocrysts is at 82.5, with a short tail to lower Mg#$_{cpx}$. The Mg#$_{cpx}$ KDE for clinopyroxene crystals from tephra samples is highly asymmetric, with a peak at 81.5 and crystal compositions extending down to Mg#$_{cpx}$ = 33.0 (Fig. 4c). Olivine crystals in the Wolf samples typically have high Mg#$_{ol}$ in the range 75.7–82.8; only two phenocryst analyses have lower Mg#$_{ol}$ (~72; Fig. 4e).

In the Fernandina nodules, independent plagioclase grains (i.e. crystals that are not inclusions) have highly variable An# (33.7–83.0), with plagioclase inclusions and crystals bounding miarolitic cavities at intermediate compositions (54.8–76.5 and 57.4–69.8, respectively). Regardless of their textural association, plagioclase An# KDEs are non-Gaussian, with their largest peaks at similar An# (67.6–68.8) and tails towards albitic compositions. Plagioclase crystals in historic Fernandina lavas have similarly diverse An# (43.5–89.7), with the largest KDE peak at 63.5. The Mg#$_{cpx}$ of clinopyroxene crystals in the 1968 Fernandina nodules and historic lavas typically range ~64–82, with the largest KDE

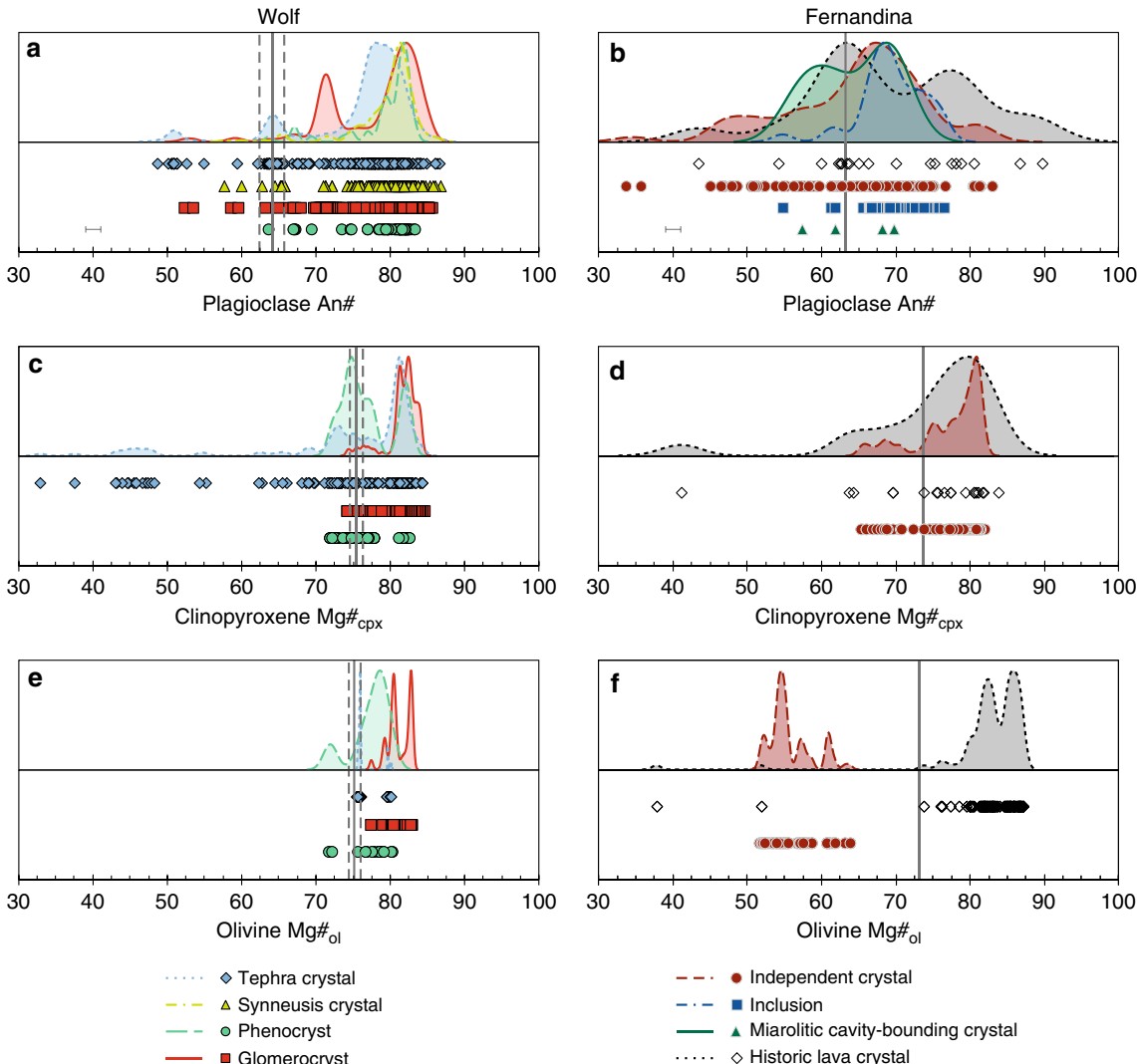

**Fig. 4 Major element compositions of minerals from Wolf and Fernandina. a**, **b** Plagioclase An#, **c**, **d** clinopyroxene Mg#$_{cpx}$ and **e**, **f** olivine Mg#$_{ol}$ in lava and tephra samples from the 2015 Wolf eruption, nodule samples from the 1968 Fernandina eruption and lava samples from historic Fernandina eruptions. Crystals are classified according to their textural association (see legend). Clinopyroxene compositions from the 2015 Wolf eruption are from Stock et al.[19] and crystal compositions from historic Fernandina lavas are from Allan and Simkin[28] and Koleszar et al.[35]. Characteristic 2$\sigma$ analytical uncertainties for our mineral analyses are either shown or are less than the size of a data point. The kernel density estimates above each panel show the distributions of the crystal compositions. The vertical grey lines show the compositions of crystals calculated to be in equilibrium with the 2015 Wolf tephra glass (solid lines—average composition; dashed lines—1$\sigma$ compositional range)[19] and 1968 Fernandina scoria glass[28]. Equilibrium calculations were performed at 1160 °C for Wolf and 1130 °C for Fernandina (the approximate pre-eruptive crystallisation temperatures)[19, 28], using the models of Namur et al.[67] for plagioclase, Putirka[68] for clinopyroxene and Herzberg and O'Hara[69] for olivine.

peaks at similar values (81.0 and 79.9, respectively; Fig. 4d). Olivine crystals in the nodule samples are stoichiometric but have low Mg#$_{ol}$ (51.7–63.8) with the largest KDE peak at 54.7. Olivine crystals in historic Fernandina lavas have been identified with very low Mg#$_{ol}$ (>37.8) but most analyses are in the range 73.9–86.7, with the largest KDE peak at 86.1 (Fig. 4f)[28,35].

Plagioclase minor element concentrations in our samples vary systematically with An#. $K_2O$ concentrations range from below detection limit (~0.02 wt%) to 0.38 wt% in the Wolf samples and from 0.12 to 1.22 wt% in the Fernandina nodules. $K_2O$ correlates negatively with An# in both eruptions, but the gradient is significantly steeper in the Fernandina samples. Plagioclase crystals from historic Fernandina lavas typically have lower $K_2O$ concentrations than in the 1968 nodules but overlap with the Wolf lavas (Fig. 5a). Plagioclase $TiO_2$ concentrations are 0.04–0.22 wt% and 0.09–0.30 wt% in the Wolf samples and

Fernandina nodules, respectively. In both cases, $TiO_2$ correlates negatively with An# at high An# but positively with An# at low An#. In Wolf samples, the inflection occurs at An# ≈63 and $TiO_2$ ≈0.19 wt%, whereas in Fernandina samples it is at lower An# (~57) and higher $TiO_2$ (~0.25 wt%; Fig. 5b). FeO concentrations are typically highest at intermediate An#, with the lowest concentrations in the most albitic and anorthitic plagioclase analyses, although there is significant scatter (Fig. 5c). Plagioclase MgO concentrations in the Wolf samples define two populations, which each correlate negatively with An#: the first has high MgO concentrations (0.08–0.28 wt%) and includes crystals in all textural associations from lava and tephra samples; the second has lower MgO concentrations (0.03–0.10 wt%) and includes only a sub-set of crystals from the tephra samples. Plagioclase crystals in the Fernandina nodules typically have lower MgO contents than the Wolf lavas; only a few analyses extend up to 0.19 wt%. In

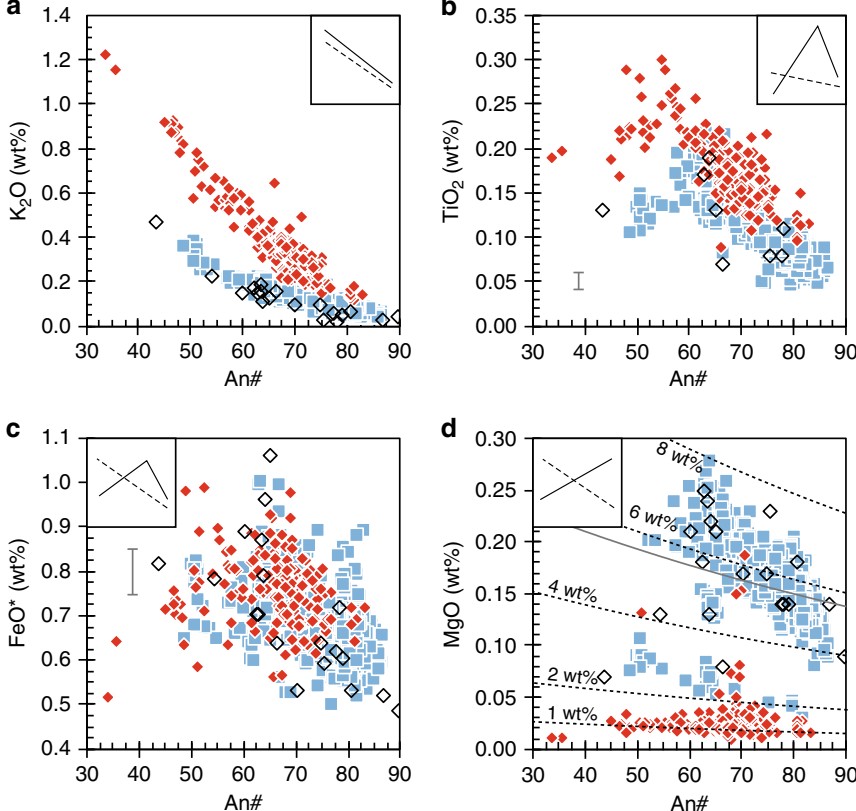

**Fig. 5 Plagioclase minor element compositions from Wolf and Fernandina.** Minor elements vs. An# of plagioclase crystals in lava and tephra samples from the 2015 Wolf eruption (blue squares), nodule samples from the 1968 Fernandina eruption (red diamonds) and lava samples from historic Fernandina eruptions (open diamonds). Crystal compositions from historic Fernandina lavas are from Allan and Simkin[28]. Characteristic $2\sigma$ analytical uncertainties for our plagioclase analyses are shown or are less than the size of a data point. The lines in (**d**) show the MgO contents of plagioclase crystals calculated to be in equilibrium with liquids containing 1, 2, 4, 6 and 8 wt% MgO (at appropriate temperatures from Wolf Rhyolite-MELTS models at 300 MPa; black dashed lines) and with the average 2015 Wolf tephra glass at 1160 °C (the approximate pre-eruptive crystallisation temperature; solid grey line)[19], using the MgO partitioning model of Nielsen et al.[43]. Insets show the general theoretical trends of crystal compositions during growth (i.e. fractional crystallisation; solid lines) and diffusive re-equilibration (dashed lines), after Humphreys[39].

contrast, most plagioclase analyses from historic Fernandina lavas have higher MgO contents that correlate with the high-MgO Wolf population (Fig. 5d).

In samples from the 2015 Wolf eruption, plagioclase crystals are typically either unzoned or show concentric oscillatory zoning. Some plagioclase glomerocrysts in lava samples have highly resorbed cores with very low An# and high $K_2O$ and MgO concentrations, overgrown by higher An# mantles (Figs. 3b, 6a). Additionally, a few crystals in the tephra samples have high-An# cores and normally zoned mantles that extend to low An# and high $K_2O$; these crystals can have anorthite contents covering almost the full range identified in our Wolf samples (Fig. 6b). Some crystals in both lava and tephra samples have thin, normally zoned rims that show a small decrease in An#, accompanied by increases in $K_2O$, MgO, FeO and $TiO_2$ (Fig. 6a); other crystals have elevated FeO at their rims even at constant An#. Zoning in clinopyroxene crystals from the 2015 Wolf eruption was described by Stock et al.[19], who identified rare patchy zoned crystals with low $Mg\#_{cpx}$ and reverse zoned crystals with highly resorbed low-$Mg\#_{cpx}$ cores in tephra samples. We have subsequently identified ilmenite inclusions within these crystals (Fig. 3d). However, most clinopyroxene crystals from the 2015 eruption are either unzoned or oscillatory zoned, occasionally with slightly higher $Mg\#_{cpx}$ cores and/or sector zoning (Fig. 3e). Olivine is typically unzoned, but some crystals show normal zoning with decreasing $Mg\#_{ol}$ towards their rims. Most

plagioclase crystals in nodule samples from the 1968 Fernandina eruption show minor oscillatory zoning in An#, $K_2O$, FeO and/or MgO, but a minority contain more significant normal and reverse zones. Many plagioclase grains have a normal zone at their rims, extending to low An# and high $K_2O$ (Fig. 6c) and FeO; these zones are typically thick (<100 μm) in isolated crystals. Clinopyroxene crystals are similarly unzoned or show slight oscillatory zoning and many have a thick (<250 μm) normally zoned rim, extending to low $Mg\#_{cpx}$. Olivine crystals in the Fernandina nodules are generally unzoned but, in contrast with other phases in our samples, some have reverse zoned rims, characterised by a small increase in $Mg\#_{ol}$.

**Evaluating crystal−liquid equilibria.** To investigate the degree of compositional heterogeneity in sub-volcanic melts, we calculated the compositions of minerals that would have been in equilibrium with their carrier liquids (represented by tephra glasses from each eruption) and compared these with the compositional distributions of our mineral analyses (Fig. 4; see "Methods"). For Wolf, the average equilibrium plagioclase An# is $64.1 \pm 1.7$ ($1\sigma$ of the range calculated from all available glass analyses), clinopyroxene $Mg\#_{cpx}$ is $75.5 \pm 0.9$ and olivine $Mg\#_{ol}$ is $75.2 \pm 0.8$. For Fernandina, the average equilibrium plagioclase An# is 63.3 ($1\sigma$ cannot be determined as only one glass analysis is available), clinopyroxene $Mg\#_{cpx}$ is 73.6 and olivine $Mg\#_{ol}$ is 73.1.

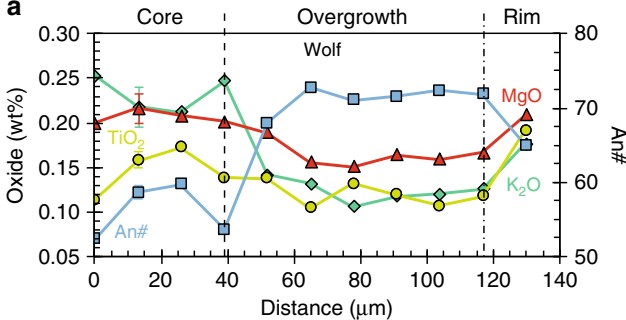

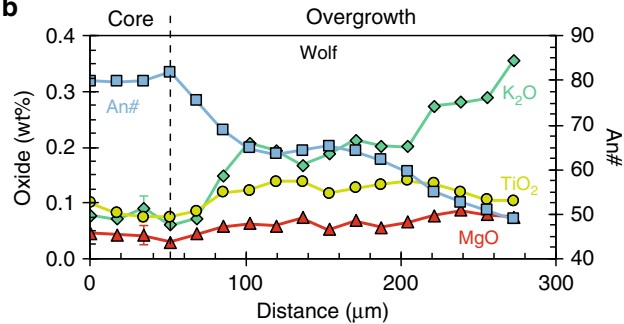

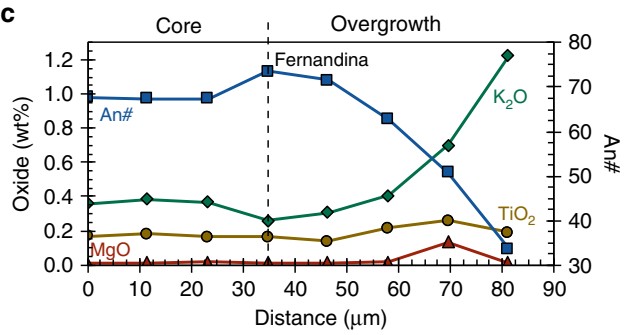

**Fig. 6 Selected plagioclase zoning profiles.** Core-to-rim zoning profiles across **a** a glomerocrystic plagioclase crystal from a 2015 Wolf lava sample, **b** a plagioclase crystal from a 2015 Wolf tephra sample, and **c** a plagioclase crystal from a 1968 Fernandina nodule sample. MgO is typically below detection limit in (**c**). Characteristic 2σ analytical uncertainties are shown or are less than the size of a data point.

Most of our mineral analyses from the 2015 Wolf and 1968 Fernandina samples have higher An# and Mg# than predicted to be in equilibrium with their carrier liquids (Fig. 4). This is typical of ocean island volcanoes globally, where magmas evolve through fractional crystallisation, precipitating minerals with progressively lower An# and Mg# until the ascending carrier liquid entrains earlier-formed crystals and brings them to the surface[36]. However, many of the mineral compositions in our samples extend to much lower An# and Mg# values than predicted to be in equilibrium with their carrier liquids. These highly evolved compositions are found in the interiors of plagioclase and clinopyroxene crystals from Wolf tephra, plagioclase glomerocrysts from Wolf lavas, and all mineral phases analysed in the Fernandina nodules. Evolved compositions have also been measured in rare olivine, plagioclase and clinopyroxene crystals from historic Fernandina lavas[28]. Typically, they form long tails in non-Gaussian An# and Mg# KDEs, suggesting that crystals grew or equilibrated with a range of variably evolved melts (Fig. 4). Hence, the crystal cargos of the 2015 and 1968 eruptions record highly heterogeneous liquids in the Wolf and Fernandina sub-volcanic systems, despite the volcanoes erupting monotonous melts at the surface over long timescales.

**Characterising equilibrium melt compositions.** Diffusive re-equilibration of An# (i.e. CaAl–NaSi interdiffusion) and highly charged cations (e.g. Ti$^{4+}$) is very slow in plagioclase, causing minerals to retain their original compositions over millennia, even at magmatic temperatures (insets Fig. 5)[37,38]. Hence, we interpret plagioclase TiO$_2$ variations in our Wolf and Fernandina samples as reflecting compositional changes in their host liquids at the time of crystallisation. By comparing models of plagioclase TiO$_2$ concentrations during fractional crystallisation with our analyses from Galápagos, we can constrain the composition of their equilibrium liquids and quantify the extent of heterogeneity in sub-volcanic melts[39–41]. Our modelling approach calculates plagioclase major element and residual liquid compositions during isobaric cooling and fractional crystallisation using Rhyolite-MELTS[42], and computes equilibrium plagioclase TiO$_2$ concentrations at each temperature step using the temperature- and anorthite-dependent partitioning model of Nielsen et al.[43] ($D_{Ti} = 0.03$–0.10; Supplementary Dataset 7; see "Methods"). Comparing natural plagioclase analyses to the Rhyolite-MELTS outputs provides the major element composition, and hence an estimate of the physical characteristics (density and viscosity), of their equilibrium liquids.

The trajectory of plagioclase TiO$_2$ vs. An# predicted by our models matches the compositional trend defined by crystals from Wolf and Fernandina (Figs. 7, 8). We ran the models over a range of pressures, guided by independent petrological and geophysical constraints on magma storage depths at Wolf and Fernandina[19,21,32,34]. The model outputs are relatively insensitive to pressure but taken at face value the best fits to the natural data are at 300 MPa for Wolf and 500 MPa for Fernandina, in agreement with previous estimates of the main pressures of magma storage[19,34]. Simulated crystal compositions underpredict the TiO$_2$ content of our very lowest An# crystals from Fernandina, potentially due to a paucity of experimental data for these compositions[43]. Both Wolf and Fernandina models accurately predict the inflection where TiO$_2$ transitions from correlating negatively with An# to correlating positively. In our models, these inflections occur when ilmenite comes onto the liquidus, suggesting that crystals with An# ≲57–63 grew from ilmenite-saturated melts, despite the absence of ilmenite as a phenocryst phase in any of the western Galápagos basaltic shield volcanoes. Ilmenite inclusions are, however, found within the most evolved (lowest Mg#$_{cpx}$) clinopyroxene cores from the 2015 Wolf eruption (Fig. 3d), validating the model predictions.

Our models suggest that the most evolved natural plagioclase crystals (i.e. lowest An#) from Wolf and Fernandina grew from plagioclase- and clinopyroxene-saturated basaltic trachy-andesitic melts with 2.46 wt% MgO (Mg#$_{liq}$ = 37.5; at 300 MPa) and trachy-andesitic melts with 1.08 wt% MgO (Mg#$_{liq}$ = 25.1; at 500 MPa), respectively (Figs. 7, 8). These melt compositions are substantially more evolved than any known eruptive rocks from Wolf or Fernandina (Fig. 2). In fact, the presence of resorbed quartz in tephra from the 2015 eruption suggests that our plagioclase data may not capture the full compositional range of sub-volcanic liquids because quartz does not saturate in our models until melts reach dacitic-to-rhyolitic compositions at low temperatures (<830 °C). Quartz is not a phenocryst or groundmass phase in the rhyodacites of Alcedo or Rabida volcanoes (or any other Galápagos lava that has been inspected)[8,31], but it is present in Rabida crustal xenoliths[44]. Hence, our data suggest that Galápagos volcanoes that have erupted monotonous basaltic lava over millennial timescales contain highly evolved, silicic melts within their sub-volcanic systems that have not previously been identified from material erupted at the surface.

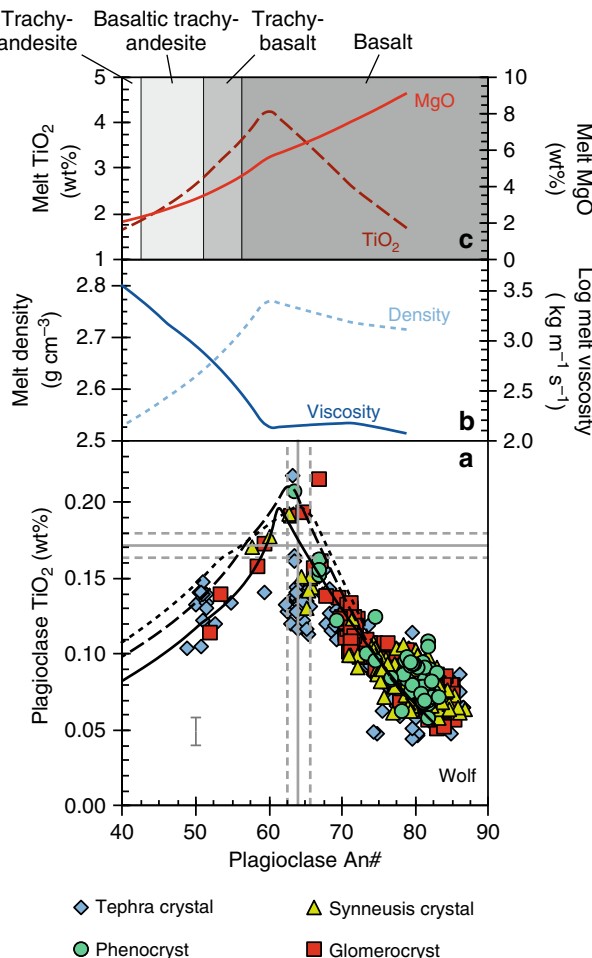

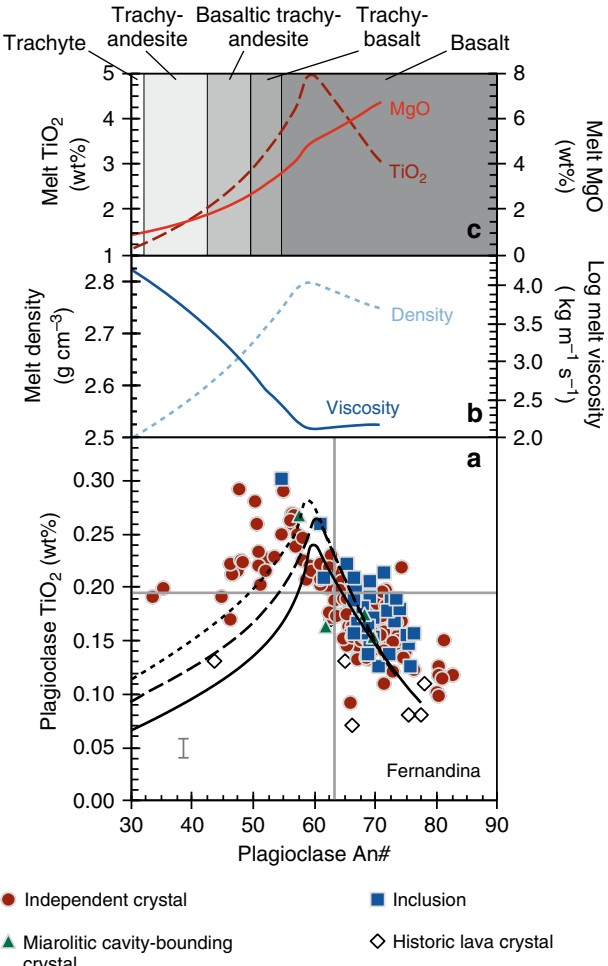

**Fig. 7 TiO₂ in plagioclase model for the 2015 Wolf eruption. a** Plagioclase
TiO₂ vs. An# in lava and tephra samples from the 2015 Wolf eruption.
Crystals are classified according to their textural association (see legend).
Characteristic 2σ analytical uncertainty is shown for TiO₂ and is less than
the size of a data point for An#. The grey lines show the compositions of
crystals calculated to be in equilibrium with the 2015 Wolf tephra glass
(solid lines—average composition; dashed lines—1σ compositional range)[19],
using the models of Namur et al.[67] for An# and Nielsen et al.[43] for TiO₂ at
1160 °C (the approximate pre-eruptive crystallisation temperature)[19]. The
black lines show the trajectory of plagioclase compositional evolution
calculated using Rhyolite-MELTS and the TiO₂ partitioning model of Nielsen
et al.[43] at 50 MPa (solid line), 300 MPa (dashed line) and 500 MPa (dotted
line). Plagioclase comes onto the liquidus at 1214, 1229 and 1244 °C in 50,
300 and 500 MPa models, respectively. The plots above show **b** the
physical and **c** the compositional evolution of the liquid predicted by
Rhyolite-MELTS at 300 MPa (the approximate crystallisation pressure of
the main magma storage zone at Wolf)[19].

**Diffusive re-equilibration**. Iron and Mg diffuse faster than Ti in
plagioclase, and initial crystal compositions for these elements are
hence more likely to be overprinted by diffusive re-
equilibration[38,39]. In our samples, Fe shows significant scatter
(Fig. 5) and is often enriched at crystal rims where An# and other
elements remain constant. We believe that these apparently high-
Fe rims likely reflect secondary fluorescence induced during
electron probe microanalysis (EPMA) and are not of geological
significance[45].

Magnesium is negligibly affected by secondary fluorescence[45]
and if the Mg content of Wolf and Fernandina plagioclase were
controlled by changes in melt chemistry during fractional
crystallisation we would anticipate a positive correlation between

**Fig. 8 TiO₂ in plagioclase model for the 1968 Fernandina eruption.**
**a** Plagioclase TiO₂ vs. An# in nodule samples from the 1968 Fernandina
eruption and lava samples from historic Fernandina eruptions. Crystals are
classified according to their textural association (see legend). Crystal
compositions from historic Fernandina lavas are from Allan and Simkin[28].
Characteristic 2σ analytical uncertainty for our plagioclase analyses is
shown for TiO₂ and is less than the size of a data point for An#. The grey
lines show the compositions of crystals calculated to be in equilibrium with
the 1968 Fernandina scoria glass[28], using the models of Namur et al.[67] for
An# and Nielsen et al.[43] for TiO₂ at 1130 °C (the approximate pre-eruptive
crystallisation temperature)[28]. The black lines show the trajectory of
plagioclase compositional evolution calculated using Rhyolite-MELTS and
the TiO₂ partitioning model of Nielsen et al.[43] at 50 MPa (solid line), 300
MPa (dashed line) and 500 MPa (dotted line). Plagioclase comes onto the
liquidus at 1191, 1207 and 1222 °C in 50, 300 and 500 MPa models,
respectively. The plots above show **b** the physical and **c** the compositional
evolution of the liquid predicted by Rhyolite-MELTS at 500 MPa (the
approximate crystallisation pressure at Fernandina)[34].

An# and MgO (inset Fig. 5c). Our data, however, define three
populations, each with nearly constant MgO or a negative
correlation between An# and MgO (Fig. 5). This relationship is
consistent with diffusive re-equilibration where MgO concentra-
tions reflect lower solid−liquid Mg partition coefficients in more
anorthitic plagioclase[39,43]. To determine the compositions of the
liquids with which these plagioclases equilibrated, we modelled
crystal compositions in equilibrium with liquids of varying MgO
concentration, using temperatures from Rhyolite-MELTS and the
MgO partitioning model of Nielsen et al.[43] ($D_{Mg}$ = 0.02–0.05).
We find that most crystals from the 1968 Fernandina nodules are

consistent with having re-equilibrated with melts containing 1–2 wt % MgO, similar to the most evolved magma predicted from our Ti modelling (Figs. 5d, 8). Although crystals from the 2015 Wolf eruption do not preserve an MgO fractional crystallisation trend, they are also not in equilibrium with any melt at fixed MgO (Fig. 5d). We interpret this as recording disequilibrium, whereby the higher MgO population have partially re-equilibrated with melts analogous to the carrier liquid (>4 wt% MgO) and the tephra crystals with lower MgO contents are partially equilibrated with more evolved melts (~2–3 wt% MgO; Fig. 5d). Some crystals contain zones with slightly elevated MgO close to their rims, potentially due to intermittent growth from more primitive melts or boundary layer effects during rapid crystallisation (e.g. Fig. 6a, c)[46].

Olivine crystals in our Fernandina nodule samples have ubiquitously low $Mg\#_{ol}$, consistent with growth or re-equilibration with evolved liquids. They are typically unzoned, but a minority of crystals have reverse rims (in contrast with normal plagioclase and clinopyroxene rim zones). As Fe–Mg interdiffusion in olivine is geologically fast[47], this could reflect either growth or diffusive re-equilibration with more primitive liquids on very short pre-eruptive timescales. Although a sub-set of plagioclase crystals in Wolf tephra samples have similarly low MgO contents, no low-$Mg\#_{ol}$ olivine has been identified in these samples.

**Storage depths and origin of Galápagos evolved liquids**. We find evidence for evolved liquids in different types of sample (lava, tephra, nodules) and crystal associations (e.g. tephra crystals, glomerocrysts) from Wolf and Fernandina volcanoes. In our Wolf lava samples, low-An# plagioclase crystals that grew from basaltic trachy-andesitic melts are in the same glomerocrystic aggregates as pyroxenes that crystallised at ~300 MPa (from clinopyroxene-melt barometry, ±140 MPa standard error of estimate)[19]; this is consistent with glomerocrysts being sourced from a magma storage region at >6.1–8.8 km [19] and provides strong evidence for evolved liquids in the lower crust. However, albitic plagioclase, low-$Mg\#_{cpx}$ pyroxenes and resorbed quartz crystals are also present in tephra from the 2015 eruption. Stock et al.[19] interpret the tephra as deriving from an upper crustal storage region (identified geophysically at ~1 km depth), based on it being the first material to have erupted and having a crystal cargo distinct from that of later lavas. Similarly, the Fernandina nodules contain evolved olivine and feldspar crystals, as well as miarolitic cavities and evidence of alteration by an active hydrothermal system, which demonstrate that they formed in an upper crustal storage region, at a similar level to a shallow geophysical source at ~1 km depth[21,48]. Hence, our data qualitatively indicate that evolved liquids occur at a variety of depths beneath Galápagos volcanoes.

Rhyolite-MELTS models show that the viscosity of Galápagos melts increases significantly after ilmenite saturation, accompanied by a reduction in density (Figs. 7b, 8b). Magma buoyancy alone is often insufficient to drive eruptions[49], and the low $H_2O$ content of Galápagos primary melts[50] would delay crystallisation-induced volatile saturation (i.e. second boiling) and the generation of volatile overpressure until very low melt fractions (e.g. refs. [51,52]). Hence, we suggest that once ilmenite-saturated magmas stall and stagnate in the crust beneath Galápagos volcanoes, they become highly viscous (due to their high melt viscosities and crystallinities) and are then unlikely to ascend further without external influence (e.g. mafic recharge)[53] or until they have undergone substantial crystallisation.

**Basalt flushing beneath monotonous volcanoes**. Although crystal compositions provide unequivocal evidence of highly variable melts in Galápagos sub-volcanic systems, such heterogeneity is not reflected in the geochemistry of monotonous lavas erupted at the surface. To determine the fate of these magmas, we studied the petrographic contexts of evolved mineral zones. Plagioclase and clinopyroxene crystals in our samples have diverse textures, with crystals from Wolf lavas and tephra including both normal and reverse zoning (Figs. 3d, e, 6a, b), and crystals from Fernandina nodules containing oscillatory normal and reverse zones, often with the lowest An# and $Mg\#_{cpx}$ at their rims. Although the scarcity of crystals with evolved zones inhibits robust characterisation of populations, these textures indicate open systems, with intermittent interactions between primitive and evolved melts[54]. Glomerocrysts in the 2015 Wolf eruption derive from disaggregated sub-volcanic mush[19] and some glomerocrystic plagioclase grains contain evolved (low An#) cores enclosed by fully concentric primitive (high An#) mantles, consistent with mafic recharge before the crystals were incorporated into a cumulate pile (i.e. while they were surrounded by melt; Fig. 3b). Mush accumulation likely occurred over long timescales and this textural evidence, along with diffusive re-equilibration of plagioclase MgO contents, suggests that some mixing events recorded in zoned crystals significantly pre-date eruption. Furthermore, as clinopyroxene-melt barometry indicates that glomerocrysts are derived from the lower crust[19], at least some of this mixing likely occurred at depth.

Given the petrographic evidence for open-system behaviour, we constructed $K_2O/TiO_2$ vs. $Mg\#_{liq}$ mixing models to determine the proportions of primitive and evolved melts in the mono-tonous lavas erupted at Wolf and Fernandina (Fig. 9). This ratio is useful because it increases dramatically over a short crystal-lisation interval after ilmenite saturation and is unaffected by plagioclase accumulation. The evolved end-member in our models is taken as the melt calculated to be in equilibrium with our lowest An# plagioclase crystals from Wolf and Fernandina (from Rhyolite-MELTS) and the primitive end-members are the highest and lowest $Mg\#_{liq}$ erupted liquid or melt inclusion from each volcano (some higher $Mg\#_{liq}$ whole-rock analyses from Fernandina are excluded as they have accumulated olivine and are not true liquids; Supplementary Table 1). Figure 9 shows that all the liquids (whole-rocks and glasses) erupted at Wolf have low $K_2O/TiO_2$ ratios, approximately along the liquid line of descent, and contain <10% of the evolved magma. Most liquids erupted at Fernandina also have low $K_2O/TiO_2$ ratios and contain <10% of the evolved end-member. However, Geist et al.[34] identified a small number of submarine lavas on the southwest flank of the volcano that have more evolved compositions and are derived from the lower crustal storage region. These evolved series lavas have a restricted range of $Mg\#_{liq}$ (~40–50), suggesting that their spatially related vents were fed by compositionally similar primitive liquids ascending from depth, but have elevated $K_2O/TiO_2$ ratios and may contain up to ~50% of the evolved magma. Our estimates of the amount of evolved material in erupted magmas are probably maxima, as quartz crystals in our tephra samples suggest that magmas fractionated beyond the extent recorded by our plagioclase analyses; real evolved end-member melts may have significantly higher $K_2O/TiO_2$ ratios than those in our models.

Our petrological data and models show that evolved melts exist in Galápagos sub-volcanic systems and periodically mix with more primitive melts. However, they typically comprise <10% of the magma erupted at the surface. We thus suggest that their absence from the erupted record is due to mass-balance, whereby large volumes of basaltic magma ascending through the crust from a primitive lower crustal storage region interact with much smaller quantities of more evolved, heterogeneous magma at higher levels (Fig. 10). The liquids mix, but because of the disparity in their relative proportions, the basaltic component dominates the mass-balance and its composition remains almost

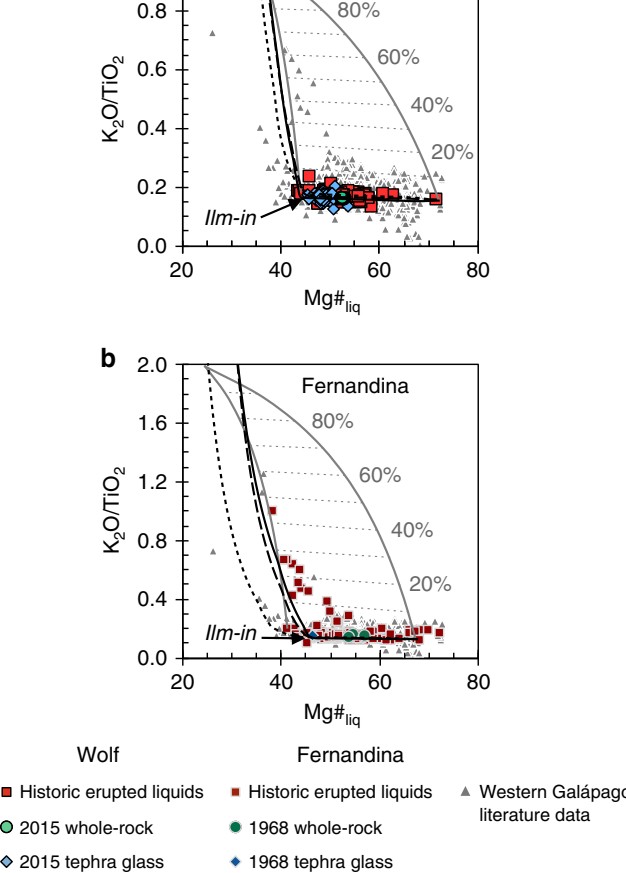

**Fig. 9 Mixing models between evolved and primitive end-member liquids.** $K_2O/TiO_2$ vs. $Mg\#_{liq}$ of erupted liquids at **a** Wolf volcano and **b** Fernandina. The $K_2O/TiO_2$ ratio of evolving liquids increases significantly when ilmenite (ilm) comes onto the liquidus. The points show: whole-rock, tephra glass, submarine glass and melt inclusion literature data from all volcanoes in the western Galápagos Archipelago (excluding intrusive rocks and plagioclase-ultraphyric lavas); historic erupted liquids (i.e. whole-rocks, tephra glasses and submarine glasses) from Wolf and Fernandina; and whole-rock and tephra glass data from the 2015 Wolf and 1968 Fernandina eruptions (see legend). References for Wolf and Fernandina historic erupted liquids and literature data from all western Galápagos volcanoes are in Supplementary Note 2. Glass data for the 2015 Wolf eruption are from Stock et al.[19] and whole-rock data are from this study. Glass and whole-rock data for the 1968 Fernandina eruption are from Allan and Simkin[28]. Characteristic $2\sigma$ analytical uncertainties for our whole-rock analyses are less than the size of a data point. The black lines show liquid lines of descent predicted by Rhyolite-MELTS at 50 MPa (solid line), 300 MPa (dashed line) and 500 MPa (dotted line). The solid grey lines show mixing models between evolved and primitive liquids, contoured by increasing proportions of the evolved end-member (dashed grey lines). For both volcanoes, the evolved (high $K_2O/TiO_2$) end-members are the liquids calculated to be in equilibrium with our lowest An# plagioclase crystals using Rhyolite-MELTS (at 300 MPa for Wolf and 500 MPa for Fernandina; Supplementary Table 1). The primitive (low $K_2O/TiO_2$) Wolf end-members are the highest (whole-rock W9562) and lowest (glass D4A) $Mg\#_{liq}$ liquids measured in historic eruptions by Geist et al.[17] The primitive Fernandina end-members are the highest (melt inclusion D25C-2-34) and lowest (glass D30-A) $Mg\#_{liq}$ normal series liquids (i.e. excluding whole-rock samples that include accumulated olivine) measured in historic eruptions by Koleszar et al.[35] and Geist et al.[34], respectively (Supplementary Table 1).

unaltered; only evolved crystals that were not fully resorbed during the mixing event preserve a record of the earlier heterogeneity. Hence, monotonous activity does not reflect simplicity or chemical homogeneity in magmatic systems. Instead, just as $CO_2$ flushing through the crust from depth can impact the composition of magmatic volatiles at higher levels[55], we suggest that monotonous volcanism reflects large volumes of basalt flushing through the crust from depth reacting with and overwhelming smaller volumes of more evolved, heterogeneous material at shallower levels (Fig. 10).

A large body of theory considers basaltic injections into silicic magma reservoirs, largely in the context of driving large-volume explosive eruptions (e.g. refs. 56,57). In contrast, the hybridised monotonous basalts of Galápagos erupt in a Hawaiian or Strombolian style, despite originating from mixing between basaltic and rhyolitic magmas. We note that the 2018 lower-rift zone eruption of Kīlauea volcano (Hawai'i) also involved injection of basalt into a silicic crustal reservoir, yet the eruption was overwhelmingly effusive[58]. The critical factor in controlling mixing dynamics must therefore be the proportion of mafic and silicic magmas[59]: in both the Galápagos and Hawai'i cases, the volumes of basalt were substantially greater than those of the resident silicic magmas.

Basalt flushing will only be able to maintain monotonous eruptions while the melt flux from the lower crust is high. When the melt flux is lower or a thermally stable lower crustal storage region is yet to develop, there may be insufficient volumes of primitive melt moving through the crust to fully overprint heterogeneity at higher levels. Model results indicate that a magma flux $>1 \cdot 10^{-4}\,km^3\,year^{-1}$ is required to stabilise a super-solidus crustal mush zone[60,61]. This less than the long-term eruptive fluxes at Wolf ($0.4–1 \cdot 10^{-3}\,km^3\,year^{-1}$)[17] and Fernandina ($4.4 \cdot 10^{-3}\,km^3\,year^{-1}$)[25], but not in the eastern Galápagos Archipelago where eruptive fluxes are several orders of magnitude lower[26]. Hence, our findings support a model[8] whereby the diversity of erupted products at Galápagos volcanoes is dictated by their position relative to the centre of the hotspot (and thus the melt flux). More broadly, our findings have implications for volcano monitoring, suggesting that even volcanoes that have reliably erupted basaltic lavas for millennia can contain evolved liquids in their sub-volcanic systems. Although basalt flushing can maintain monotonous eruptions over long timescales, external influences (e.g. changes in the regional stress field) or a decrease in the crustal melt flux (e.g. resulting in further fractionation and an increased likelihood of second boiling) might allow these melts to ascend, generating explosive silicic eruptions.

## Methods

**Sample selection and preparation.** Samples from the 2015 Wolf eruption include the lava and reticulitic tephra analysed by Stock et al.[19], plus additional lava samples from the east flank of the volcano collected during fieldwork in June 2017 (Fig. 1b). All the material was unaltered and lava samples were collected from dense flow interiors where possible. Samples from the 1968 Fernandina eruption are gabbroic nodules collected on the floor and rim of the caldera during fieldwork in July 1970 (Fig. 1c) and are described by Howard et al.[48]. Most of the nodules are fresh, but some show extensive hydrothermal modification[48]. The Wolf lava samples and Fernandina nodule samples were prepared as polished thin sections. Crystals in the Wolf tephra samples were separated from the 40–500 μm size fraction by heavy liquid and magnetic separation and mounted in epoxy along with the quenched glass.

**Analytical methods.** Whole-rock samples from the 2015 Wolf eruption were analysed by X-ray fluorescence spectrometry (XRF) for major and trace elements. Analyses were performed using a Philips PW 2404 instrument at the University of Edinburgh (UK) following the method of Fitton et al.[62] with modifications by Fitton and Godard[63]. Analytical precision, encompassing errors associated with sample preparation and heterogeneity, was estimated by preparing and analysing three replicates of the same sample. Relative $1\sigma$ precision is better than ±1% for major elements (>1 wt%) and better than ±2% for minor and trace elements (<1 wt%), except Th (±8%) and Pb (±20%).

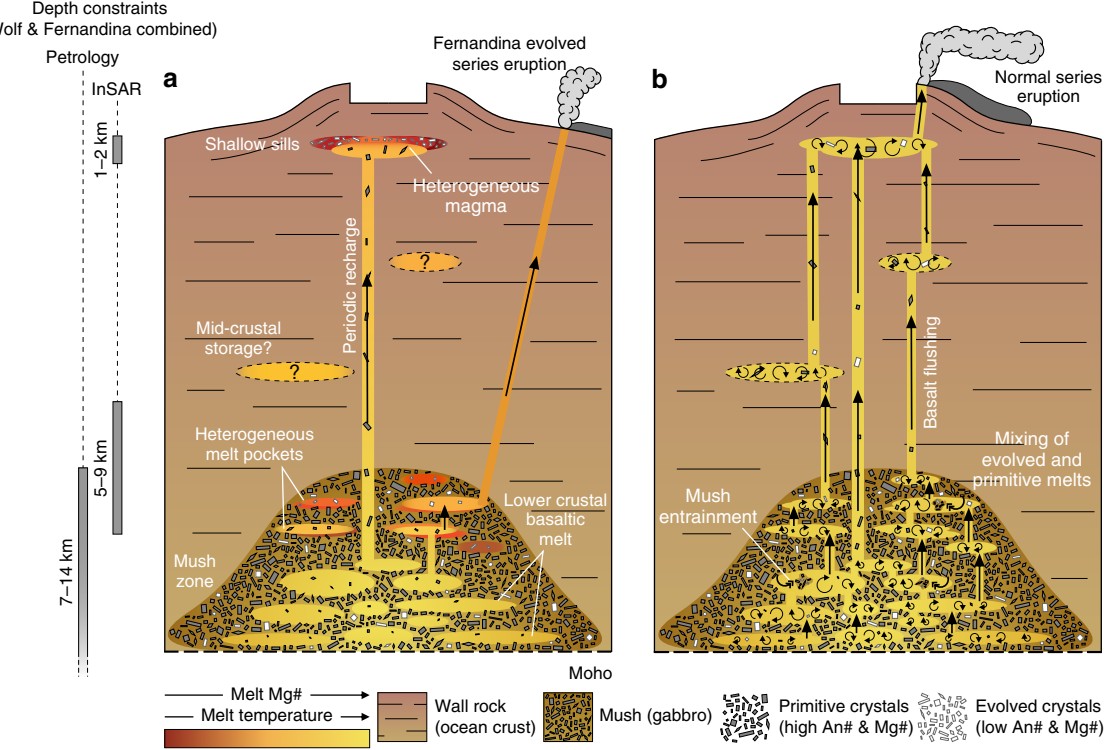

**Fig. 10 Cartoon summarising the architecture of the Wolf and Fernandina plumbing systems.** The grey bars (left) show relative estimates of magma storage depths at Wolf and Fernandina from Interferometric Synthetic Aperture Radar (InSAR) inversions and petrological barometry[19,21,32,33]. Not to scale. **a** The volcanoes are underlain by large lower crustal magma storage regions and smaller shallow storage regions, which are recharged by new magma ascending from depth. Both regions contain compositionally heterogeneous melts. We cannot discount mid-crustal magma storage but there is no petrological or geophysical evidence from recent eruptions. At Fernandina, evolved series submarine eruptions are periodically fed by melts ascending directly from the deeper storage region. **b** Large volumes of basaltic melt periodically ascend from depth, flushing through the systems and mixing with liquids stored at shallower levels. Some sub-volcanic mush is entrained into the ascending liquids. As the volume of ascending basalt is much greater than the volume of evolved material at shallower levels, its composition remains almost unaltered; only crystals that were not fully resorbed during mixing preserve evidence of the earlier heterogeneity.

Mineral compositions were measured by EPMA using Cameca SX100 (for Wolf samples) and Cameca SXFive (for Fernandina samples) instruments in the Departments of Earth Sciences at the University of Cambridge (UK) and Syracuse University (USA), respectively. To ensure consistency across instruments and multiple analytical sessions, measurements were internally calibrated using appropriate Smithsonian Microbeam Standards[64]. Relative $1\sigma$ precision was monitored by repeat analysis of mineral standards on the Cambridge instrument and is assumed to be similar on the Syracuse instrument. This is better than ±2% for major elements and ±15% for minor elements, except MnO in clinopyroxene (±17%). Typical $1\sigma$ relative errors for each element analysed by XRF and EPMA are provided in Supplementary Datasets 1–6 and full details of the analytical methods are provided in Supplementary Note 3.

We assume a melt $Fe^{2+}/Fe^*$ ($Fe^* = Fe^{2+} + Fe^{3+}$) value of 0.85 when calculating $Mg\#_{liq}$. This equates to an oxygen fugacity ($fO_2$) near the quartz-fayalite-magnetite (QFM) buffer, as measured in Galápagos lavas by Peterson et al.[65]. We do not assume an Fe speciation for minerals, instead calculating the number of $Fe^{2+}$ and $Fe^{3+}$ ions per formula unit by stoichiometry and using the sum of these ($Fe^*$) in our calculations of $Mg\#_{cpx}$ and $Mg\#_{ol}$. We evaluate the distribution of our major element datasets using KDEs, with bandwidths calculated after Sheather and Jones[66].

**Modelling crystal–liquid equilibria.** Equilibrium plagioclase, clinopyroxene and olivine compositions were calculated using the models of Namur et al.[67], Putirka[68], and Herzberg and O'Hara[69], respectively, given pre-eruptive storage temperatures of 1160 °C for Wolf and 1130 °C for Fernandina[19,28]. The model of Namur et al.[67] is calibrated for anhydrous melt compositions; $H_2O$ in the melt would increase the equilibrium An# but this effect is assumed to be negligible as Galápagos magmas have consistently low $H_2O$ contents (typically < 1 wt%)[35,50].

**Modelling TiO₂ in plagioclase.** Plagioclase major element and residual liquid compositions were calculated during isobaric cooling and fractional crystallisation using Rhyolite-MELTS[42]. We use primitive whole-rock (W9562)[17] and melt inclusion (D25C-2-34)[35] compositions as our Wolf and Fernandina model starting

liquids, respectively. Equilibrium plagioclase $TiO_2$ concentrations were then calculated at each temperature step using the partitioning model of Nielsen et al.[43] (Supplementary Dataset 7). By comparing modelled plagioclase An# and $TiO_2$ concentrations with our natural mineral analyses, we can characterise their equilibrium melt compositions. This method of characterising mineral-melt relationships is preferable to that of Scruggs and Putirka[70] where the eruptive record does not encompass the full range of melt compositions in a sub-volcanic system.

Models were run at 50, 300 and 500 MPa, an $fO_2$ at the QFM buffer and 0.15 wt% initial $H_2O$, which approximate the range of Galápagos magma storage conditions identified in previous studies[19,27,28,50,65]. The liquid lines of descent predicted by Rhyolite-MELTS fractional crystallisation models are generally a good fit to previously analysed whole-rock and glass data from Wolf and Fernandina (Supplementary Figs. 1, 2) and simulated clinopyroxene compositions also match natural crystal compositions (Supplementary Fig. 3). This validates our choice of intrinsic variables and suggests that assimilation of compositionally distinct wall rock material has a negligible impact on the compositional evolution of Galápagos magmas, in agreement with previous studies[27].

## Data availability
All compositional data collected in this study are available in the supplementary datasets.

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

## Acknowledgements

M.J.S. was supported by a Charles Darwin and Galápagos Islands Junior Research Fellowship at Christ's College, Cambridge. D.G.'s effort is based upon work while serving at the National Science Foundation and was initially funded by NSF grant EAR-1145271. D.A.N. was supported by a Presidential Fellowship at the University of Manchester and M.L.M.G. was supported by an NERC studentship (NE/L002507/1). Additional fieldwork funding was provided by the Jeremy Willson Charitable Trust (administered by the Geological Society of London) and the Mineralogical Society of Great Britain and Ireland. Maps in Fig. 1 were created using JAXA ALOS imagery from http://www.eorc.jaxa.jp/. We thank Sally Gibson for her assistance with fieldwork and encouragement with this manuscript. We also acknowledge fieldwork support from the Charles Darwin Foundation, Galápagos National Park, Instituto Geofísico Escuela Politécnica Nacional and the crew of *MV Pirata*, in particular Wilson Villamar and Antonio Proaño, without whom this work would not have been possible. We thank Roel van Elsas for mineral separation, and Nic Odling and William Nachlas for technical assistance with XRF and EPMA analysis, respectively.

## Author contributions

M.J.S. conceived the project with D.G., D.A.N. and M.L.M.G.; M.J.S., D.G., M.L.M.G., B.B., and K.A.H. collected the samples; M.J.S. prepared the samples for XRF analysis and M.J.S., D.G. and I.B. prepared and analysed the samples by EPMA; M.J.S. performed the modelling and analysed the data with help from D.G., D.A.N., M.L.M.G. and J.M.; M.J.S. wrote the first draft of the manuscript, which was revised by all authors.

## Competing interests

The authors declare no competing interests.
