## [Peer Review File · Nature Communications]

REVIEWER COMMENTS

Reviewer #1 (Remarks to the Author):

Dear Associate Editor – Dr. Sebastian Mueller,
thank you for your invitation to review the manuscript entitled "Cryptic evolved melts beneath monotonous basaltic shield volcanoes in the Galápagos Archipelago" by Dr. Stock and co-authors.

In this work, the authors combine petrological data on natural phenocrysts from compositionally monotonous eruptions at Galapagos with thermodynamic modeling based on the MELTS code, in order to investigate cryptic magma differentiation processes at depth. The main conclusions are that mineral compositional changes represent important tracers for the balance between magma differentiation and recharge processes in sub-volcanic systems. In this respect, eruptions at Galapagos are persistently buffered to a basaltic composition by volumetrically significant inputs of new primitive magmas mixing with much smaller quantities (< 10 vol.%) of more evolved melts stored in shallow reservoirs.

I think that the work done by the authors represents an advancement in understanding cryptic magma dynamics beneath volcanic systems active over timescales ranging from decades to millennia. The methodological approach presented in the manuscript is noteworthy, technically sound, and well-placed in the previous literature. Overall, I have greatly appreciated this excellent work and I will be happy to see it published in Nature Communications. I have only some minor comments that can help to improve a little bit the clarity of the text and figures, making the final published paper more persuasive for the scientific community (please see also the annotated pdf uploaded to the author's webpage).

Sincerely,
Silvio Mollo

Comments:

l. 80. This entire Section refers to samples and petrography. Therefore, I would focus the related Fig. 3 only on mineral textures and compositions, avoiding the presentation of different data, such as the barometric estimates. This information can be presented only in the text.

l. 105. It is not clear how Mg#_{liq} has been calculated with respect to the iron speciation. Is total Fe expressed as Fe²⁺? There is also confusion with l. 410 stating that melt Fe²⁺/Fe* = 0.85. Is this redox state taken into account only for mineral-melt equilibria or also for the calculation of Mg#_{liq}?

l. 129. Crystal-melt equilibria have been evaluated and presented without explaining before in the text the intra-crystal compositional changes. It is therefore not clear at this stage whether crystal are chemically zoned or not and how their compositions change from core to rim. As a consequence, the statement reported at l. 175 concerning the equilibration of crystals with compositionally distinct melts, appears quite confusing because crystal-melt equilibria has been treated without informing the reader about the crystal zonation. I understand that this makes easiest the methodological approach and I agree with the strategy adopted by the authors. However, I think that it is more logic to present the mineralogical aspects earlier in the manuscript. One possibility is to discuss the mineral chemistry after the Section entitled "Samples and Petrography". In this new Section you could present most of the compositional changes of minerals reported from l. 180 to l. 220.

l. 226-236. I have greatly appreciated this modeling approach. However, I would interpolate (to some extent) the compositional change of plagioclase with the chemistry of olivine and clinopyroxene. You correctly explain that olivine may suffer for re-equilibration phenomena with compositionally distinct melts. However, I would test whether MELTS simulations may reproduce

natural clinopyroxene compositions. I would also report in the text if MELTS modeling indicates clinopyroxene + plagioclase saturation at certain P-T-H₂O conditions and if your reconstructed melt compositions are in equilibrium with both plagioclase and clinopyroxene. It is also apparent that most of your discussions and conclusions are based on MELTS simulations. For this reason, I would recommend to provide to the reader your MELTS data as supplementary material, possibly with the temperature-dependent plagioclase-melt partition coefficients calculated with the model of Nielsen et al. (2017). I agree that the incorporation of Ti in plagioclase may be a powerful tool for unraveling cryptic magma dynamics, but I think that your entire approach would appear more robust by combining a little bit different textural and chemical evidences from olivine, clinopyroxene and plagioclase.

I. 348-360. This is an excellent example of a well-developed scientific discussion.

Fig. 3. I suggest to focus the panel (a) only on mineral texture and chemistry. Please identify the plagioclase in panel (b). Please provide An# and Mg#cpx for the numbers reported on the images.

Fig. 7. Please add (a), (b) and (c) in the figure caption.

Fig. 8. The same as Fig. 7.

Fig. 9. Please argument in the caption the compositions of the two melt end-members or, alternatively, report these compositions as supplementary material.

Fig. 10. It would be great to see in this cartoon some indications on P, T, depth (in km) and, perhaps, melt-H₂O contents, if you can easily retrieve these information from previous literature studies.

I would not use kbar but rather MPa for the pressure estimates. kbar and Kelvin units are specific to thermodynamic studies calibrating models. Rather, the final target of your work is to understand magma dynamics.

Reviewer #2 (Remarks to the Author):

“Cryptic evolved melts beneath monotonous basaltic shield volcanoes in the Galapagos Archipelago” by Michael Stock et al. is an interesting and well written article, based on high quality mineral data. The authors present the case-studies of Wolf and Fernandina volcanoes which are shown to erupt basaltic magmas of striking compositional similarity. However, based on the diversity of mineral compositions and Rhyolite-MELTS simulations the authors indicate that despite the bulk compositional homogeneity of the erupted magmas, the magmatic systems also contain rhyolitic subvolcanic reservoirs. The main conclusion of the paper is that the compositionally monotonous character of the eruptions is caused by a high-influx of mafic recharge melt from the lower crust, totaling ~ 90 vol% of the erupted materials. Despite this, the volcanic system is developing a rhyolitic reservoir that could increase the risk of energetic explosive eruptions in the future.

I find this work relevant from at least two points of view:

(1) Firstly, it showcases what could very well be an initial stage of magmatic system evolution in island volcanoes worldwide, whether they are related to plume or arc magmatism. Initial high influxes of mafic recharge result in the formation of mainly basaltic edifices, while buffering the development of rhyolitic subvolcanic reservoirs. As suggested by the authors, a diminishing in the rate of mafic recharge with time might allow the development of the more silicic parts of the system, which can therefore contribute more in terms of melt, crystals and volatiles to future eruptions. We have observed a similar case while studying Methana volcano in the South Aegean Arc (Popa et al., in review, JVGR), where high influxes of mafic recharge interact with rhyolitic

subvolcanic reservoirs to produce eruptions of compositionally intermediate magmas, volumetrically dominated by the recharge material (~ 70 vol%). Similarly to the cases of Wolf and Fernandina, the rhyolitic magmas do not erupt on their own. Methana could be in a similar, albeit more "evolved" stage of evolution than the Galapagos volcanoes, potentially due to lower influxes of mafic recharge. At the other end of the spectrum we might find island volcanoes which developed large rhyolitic systems, while their earlier stages of activity are dominated by the eruption of mafic magmas (e.g. Nisyros, South Aegean Arc). I have offered the examples of Methana and Nisyros volcanoes to showcase the global relevance of the findings presented in the Stock et al. paper.

(2) Secondly, the paper highlights the possibility of transition to eruptions involving colder, water-rich and more viscous magmas which have a higher explosive potential.

I find the work to be convincing, supported by high-quality data and, as argued above, of importance to the volcanology and igneous petrology community. I have only a few thoughts and minor comments to share with the authors:

Line 47: I suggest using "constant range of temperature and composition" rather than "constant".

Line 104: extending 'the' database

Lines 136-137: I suggest mentioning here that the range in An# is recorded as zonation within individual crystals (based on Fig. 6), other than between distinct crystals.

Line 309: it is debatable whether the generation of volatile overpressure is sufficient to trigger an eruption, I would suggest avoiding the mention here. Overpressure by second-boiling actually increases the volatile saturation limit of the melt, which leads to the exsolved volatiles partly re-dissolving in the melt and a natural buffer situation is reached. The authors can check the recent work of Townsend et al 2019 GGG (Magma Chamber Growth During Intercaldera Periods), specifically fig 2b for this effect. This paper also shows that exsolution of the MVP corresponds to a sharp decrease in the eruption frequency, similarly to the work of Degruyter et al 2017 GGG (Influence of Exsolved Volatiles on Reheating Silicic Magmas) and Popa et al 2019 JVGR (A connection between magma chamber processes and eruptive styles revealed at Nisyros-Yali volcano). Unzipping by magic recharge is probably the only way upper-crustal reservoirs erupt, including in water-supersaturated rhyolitic systems (e.g. Popa et al., 2019 JVGR).

Lines 378-380: It is linked to the comment above. There is additional complexity related to the effect of second boiling during magmatic storage and eruptive styles, and it does not necessarily lead to explosive events. Water-supersaturated magmatic reservoirs can sometimes trigger explosive events, but recent work has shown that volatile-supersaturation at storage pressure will more often favor effusivity. Volatile supersaturation creates counter-intuitive feedbacks related to increasing the volatile saturation limit of the melts and increasing the bulk-compressibility of the magmatic reservoir, which allows higher volumes of recharge melt to be accommodated before a state of critical overpressure is reached. This in turn allows for efficient reheating and melt viscosity drop, which favors outgassing. Moreover, once the eruption is triggered nucleated volatiles are already present at the base of the conduit. Previously re-dissolved MVPs (under disequilibrium conditions caused by the overpressure of mafic recharge being accommodated in the upper-crustal reservoir) re-nucleate on-masse creating a foamy melt at the onset of conduit ascent. The presence of the foam combines with the reheating and viscosity reduction to favor conduit outgassing, which can neutralize the explosive potential of the magma. For more details, the authors can check the papers mentioned above (Degruyter et al., 2017; Popa et al., 2019, Townsend et al., 2019).

Figure 3 caption, line 637: I think it is (c,d,e) rather than (d,e,f)

Figure 7 and 8 captions: are a bit confusing. Plagioclase TiO₂ vs An# appear in plots (c) on both figures, while they are announced at the beginning of the captions. Maybe TiO₂ vs An# should be plots (a) instead? Also, the figure captions do not distinguish between plots a,b,c.

I hope my suggestions are useful.

Best regards,
Razvan-Gabriel Popa, ETH Zurich

Reviewer #3 (Remarks to the Author):

Review of Stock et al.
By Keith Putirka
4/3/20

Overview,

This is a well-written and well thought out study and is definitely worthy of publication. The authors present a compelling case that the monotonous magmas erupted at Wolf and Fernandina are not necessarily representative of the variety of magmas that exist in the magma plumbing system. This bolsters the view of Geist et al. (2005) that the magma mush system acts as a filter against the eruption of a wider array of magma compositions, perhaps because the mush is held always above its solidus, over the lifetime of the volcano. The variety in crystal compositions clearly attests to the existence of un-erupted evolved melts. The one thing I was left curious about is what kinds of magma supply rates are needed (thermally) to sustain the mush system above its solidus, and (compositionally) to flush the system with enough basalt to keep the felsic:mafic mixing ratios low. The authors need not conduct their own modeling, but I'm wondering if any of studies such as Annen (2008), Schopa and Annen (2013), Karakas and Dufek (2015), or Jackson et al. (2018), etc., lead to any insights about plausible Galapagos melt supply rates. I know these numerical studies are mostly focused on the generation and maintenance of silicic systems, but perhaps they are still useful in the present case. I think the discussion would be aided by some mentioning of supply rates.

The only other issue, and it's a minor one as I know this is not a key aspect of the study, is that the pressures of magma storage are probably not well constrained by the methods employed. The key assumption is that any particular whole rock trend represents an isobaric one. And MELTS or other models can provide such a P. But that inferred P may have little connection to the conditions of crystallization (it need not even be an average pressure). And yet the authors have clinopyroxene, so it seems that it should be possible to estimate at least rough crystallization depths (P estimates are always rough), which based on other studies of oceanic islands systems, may be much more polybaric than indicated here (see Klugel and Klein 2005 at Madeira; Putirka 1996 at Hawaii). Here are the challenges as I see them, and a potential solution:

1. I am not sure that Ti-in-plag is sensitive enough to P to provide a useful barometer. I may be wrong, but P is very difficult to estimate using any condensed-phase equilibria, and I think estimates are almost as good as useless in the absence of plots of P(measured) vs. P(calculated) that indicate error for the method in play (where "measured" is for experimental systems, where P is known). I don't see those kinds of tests in Nielsen et al. (2017), and when I look at Nielsen et al.'s Table 6, I don't even see a P coefficient for D(Ti), which would mean that even the D is perhaps not all that sensitive to P; but perhaps I'm just missing something.
2. The only other possible "barometer" at work is implicit in the MELTS simulation of ilmenite saturation. But MELTS is only calculating a very small apparent change in phase appearance (e.g., based on the shift in the peak in the Ti v An curve in Figs 7-8), and even that peak is probably not

a useful barometer, if one were brave enough to plot P(measured) vs. P(calculated).

3. The MELTS curves collapse onto one another at the low K₂O/TiO₂ ratios that characterize Wolf, and they miss the higher K₂O/TiO₂ parts of the Fernandina array, which the authors convincingly attribute to magma mixing. But this mixing also means that if we plotted all of the major oxides we would likely find that the MELTS-derived model liquids do not match the observed whole rock trends, which they must, since the whole rocks are mostly liquid.

The problem can be solved by using observed whole rock compositions - and the extension of such trends - as a model for plausible liquids, as I note in the comments below. If we are not dealing with antecrysts, then the liquids relevant to the mineral compositions must either fall on the whole rock trends, or on their extensions to lower or higher Mg#. By using mixing, or fractional crystallization or AFC, etc. - whatever it takes to describe and extrapolate the trends - the authors can use such trends to see if any composition along such can explain the observed mineral compositions. If the authors are interested, they can see the method in play in Scroggs and Putirka (2018). In any case, here, it looks like mixing is a dominant process and, this, not MELTS, would be the most appropriate way to estimate liquid compositions for observed mineral phases. Precisely how the curve is created, though, does not matter (except in the extrapolated regions), so long as the whole rock trend is reproduced. The method fails for antecrysts, but antecrysts can be identified by using multiple tests of equilibrium. Taking cpx as an example, an antecryst might satisfy Fe-Mg exchange equilibrium for a putative liquid, but then fail to match predicted equilibrium values for DiHd, EnFs, etc. This method is not fool-proof. We can never be sure that the crystals that pass such tests are phenocrysts. They could be antecrysts from an older event, if the liquids from an earlier episode formed along the same computed liquid trend. But in such a case the information is still useful, as to produce the same crystals, the system would have to crystallize at the same P-T conditions.

Such an approach may well yield a narrow range of pressures for the Wolf and Fernandina systems. But if I were to be money, it would be that the crystals reveal a wide, rather than a narrow range of pressures. But in any case, I don't think their pressure estimates are the main focus of this work. They have a very good case for heterogeneous liquids, and I think they support the mush-filtering concept; if the authors agree that the above arguments have some merit, they can make some tests to consider a more polybaric system. But I don't know that such an analysis should be required, though in the absence of tests, they can just say something like "If we take the Ti partitioning data and MELTS modeling at face value, then it would appear that pressures of crystallization are in the range 2.3-3 kbar...".

I hope these comments are helpful. Below are line-by-line comments.

Keith Putirka

Line-by-line comments

Line 38: "...distinct batches of magma interact..."; this model will not be familiar to most readers. Expand by noting that the magmas "...interact with gabbroic wall rock". Also, in reading Bedard (1993) and also Geist et al (2005), I think that the Geist paper, and/or Sinton and Detrick (1992), would be a better citation. To my reading, those papers make a clearer case for how eruptive monotony might be imposed on a system that maintains a crustal mush above its solidus. I don't think that monotony is an obvious result of assimilation or reaction with gabbros; but maintenance of a mush above its solidus seems plausible, and probably connects to some critical, and perhaps also near-constant, melt supply rate...?

Lines 39-40: Do you mean, here, the putative density "filter", where buoyancy controls whether (or disallows) primitive magmas move to the surface? As I am sure the authors well know, Stolper and Walker (1980) showed that MORB appeared to populate a density minimum, and they argued

that that density corresponded to that imposed by the density of rocks near the Moho. Glazner (1995) made an analogous argument for continental magmatic rocks, but arguing that the middle crust was the key density filter. Using newer models for magma density, I don't see a density minimum (see Figure 3 in Putirka, 2017; Elements). So I would argue that there is a filter, but only against 'primitiveness', not against diversity. I think this is what Bedard is arguing for also – that gabbro-melt reactions provide a filter against "primary" magmas, but a filter that does not preclude the eruption of diverse, non-primitive magmas.

Line 40: Is the argument here that some volcanoes might erupt homogeneous magmas due to differentiating at similar P-T conditions? The argument also implies that they differentiate to the same extent. So it's the critical density argument of Stolper and Walker (1980), right?

Lines 47-48: Wouldn't constant melt input – a constant melt supply rate, after some initial start-up, lead to a near "exact balance" of heat input and output?

Lines 97-98: "with clinopyroxene lining many of the void walls". Does this imply that at least some cpx grains precipitated from a vapor phase? Or only that the vapor-containing voids formed from a cpx matrix? It seems like the former is the case. Out of pure curiosity, having nothing to do with the issues here, I would be curious to know if these cpx are different or distinct in composition in any particular way.

Line 104: insert "the" before "Harpp".

Line 109: and accumulated plag also affects whole rock MgO, and accumulated olivine affects Mg#, right?

Lines 114-115: Perhaps a callout to Fig. 3 here, and for comparison, perhaps a light, dashed line in the histogram in Fig. 3 could indicate the range in Mg# from another Galapagos volcano, that is not monotonous, since monotony is to some extent, relative?

Line 119: cite standard deviation of Mg#; also cite, avg. \pm std dev for tephra.

Line 121: cite avg \pm std dev. For scoria glass.

Line 158: do you mean "heterogeneity of sub-volcanic melts"?

Lines 158-160. Why not go the other direction, as that would get more directly at your stated goal? See Scruggs and Putirka (2018; Am Min). The method is simple, and likely much more accurate. The basic idea is that

1) you use observed whole rock and glass compositions, and a mixing line (or a fractionation curve) to interpolate or extrapolate from these. (Yes, whole rocks are not liquid, but they mix with liquids; most lavas have <20% crystals so they are 80% liquid; so, liquids must lie along any mixing or differentiation trend that contains whole rock compositions).

2) Now, find out if any liquids along such a trend can satisfy tests of equilibrium for your observed mineral compositions. These are the liquids that existed in the sub-volcanic system.

Lines 161-164: If these calculated average mineral compositions are meant to be compared to the averaged noted in the prior paragraph, most of this text and the prior paragraph can be deleted - substituted by and more effectively presented in a table.

Lines 203-205: some of these patterns are intriguing, such as a "small decrease in An" accompanied by increases in K and Mg. I would think that An and Mg would be positively correlated, and that increases towards the rim would indicate recharge. But recharge should cause K to drop. Could the inverse of the anticipated trends be due to an increase in growth rate, where K and Mg just can't get out of the way fast enough as the crystal/liquid interface moves rapidly

towards the liquid?

Line 214 "...in which An# correlates negatively with K₂O." That is expected, right, for "normal zoning"?

Line 222: Perhaps cite a possible T here? I've done some calculations recently using Grove et al. (1984) and the rates are very slow; it seems that once plag has crystallized, it won't really re-equilibrate except at the sub-micron scale.

Lines 224-226. I agree, but I think you can delete all of lines 222-226 and then at the end cite Grove et al. (1984) and the new Cherniak and Watson paper at the end of the sentence.

Lines 231-232: This is not bad, but the advantage of the method that I suggest (see comments for lines 158-160) is that you don't have to worry about whether your trends are produced by fractional crystallization, or mixing or assimilation, or affected by recharge, etc. Here, by relying on MELTS, your calculated liquids are only accurate to the extent that the range of liquids that produced your observed plagioclase crystals also formed by pure fractional crystallization. For moderately compatible elements like Mg or Fe, mixing and FC are both pretty linear. But for incompatible elements there is a lot of curvature to the FC trend, and so the calculated liquids may lie quite far from the actual rocks, if compositional trends are affected by magma mixing. If you instead start with observed whole rock trends (which liquids must follow) and then ask what compositions along these trends – if any – can explain my observed mineral compositions? Then you eliminate any assumptions about the processes by which the whole rocks/liquids were generated. This method only fails if the liquids that generated your observed minerals are not on the whole rock trend. But that can only be the case for those crystals that are not part of the evolutionary suite you are examining (e.g., the method fails if you have xenocrysts, rather than autocrysts). You won't discover that through MELTS, but you can often detect that error if mineral-melt equilibrium tests do not provide consistent results between clinopyroxene and plagioclase. (Mineral-scale isotopes will also reveal the problem, but that's an expensive and time-intensive approach).

Lines 238-239: you have pyroxenes, so why use MELTS to guess at pressure when you can calculate it? Then run Rhyolite MELTS at the calculated pressures for a better estimate of the plagioclase compositions?

Lines 241-242: Again, you are better off using Ti partition coefficients for plag and then figuring out where along your whole rock trends that you can find a liquid that has the right Ti content to explain the low Ti you see in certain plag. GERM cites several studies that place the K_d for Ti between plag and melt at about 0.04, and no lower. So if this is the lowest reasonable K_d, then what Ti is needed in the liquid to explain your Ti-in-plag values? And do any of your whole rock trends either contain, or extrapolate to such values? If they do, then you have a possible liquid from which these plag crystallized from. If the Ca#(liq) also works out to give you the right An content of the plag, then you're good - you know the liquid rather precisely.

Lines 253-256: I think the very low An contents seen at Fernandina (An₃₀s) could very well be co-saturated with quartz. We've done modeling at Chaos Crags where we have An contents that reach down to about An₃₃ or so, and we've found that it's almost impossible to create these in the absence of melts that are about 70% SiO₂ or higher. We discovered this by the kind of reverse modeling I've noted above (use the crystal and whole rock trends to discover the equilibrium liquid) and the inferences have lately been confirmed by analyses of melt inclusions and matrix glass, which have the predicted compositions from our low-An data, and are quartz saturated. Of course, the analogy is imperfect, as we used Lassen-derived whole rock trends to make our predictions of the evolved (and un-erupted) liquid compositions (the recent eruptions at Lassen do not involve rhyolitic whole rocks; we infer that rhyolitic magmas exist due to the presence of the

low-An plag; this is new work but touched on in Scruggs and Putirka).

Line 272: So Mg partitioning into Plag is inversely proportion to An content? Wow. I would have guessed the opposite. I don't see a clear statement of such in the Nielsen et al paper. Could you give partition coefficients for a Ca- and Na-rich case so we can see what sorts of magnitudes are active here.

Line 276: "...melts containing 1-2 wt. % MgO". OK. But if there is magma mixing or wall rock assimilation, you might get a different answer. The MELTS model is only giving you only the liquid trajectory for a highly idealized fractional crystallization case.

Line 295: how do you get these pressures (2.5-3 kbar)? That's pretty high precision. You think you have the depths of equilibration to within a 0.5 kbar range? Perhaps this is from an average, where you are using a standard error? And of course, the method matters. Not all models have the same precision.

Line 309: This is a minor point, given modern usage, but while I think I've used the term before, I am going to avoid using "second boiling", given what I have since discovered. This term originated with theoretical studies of phase relationships in 2-component systems by Roozeboom (1901), which I think includes some experimental verification in the system KNO₃-H₂O. The case was shown that the triple points of two substances can connect to form a curve that in T-P space exhibits a maximum pressure. Upon heating at constant pressure, it would then be possible to intersect the liquid/vapor (boiling) curve twice, hence the term second boiling. But the processes we are discussing do not involve the connection of two triple points and the double intersection of a liquid/vapor curve.

I say that this is a minor point because the original phenomenon is so rarely discussed in any context, and the original usage of "second boiling" now so heavily buried in the literature, that it probably doesn't matter at this point how the phrase is used. We are perhaps free to be use it any way we like.

Lines 350-360: this all makes perfect sense. I just am not sure whether the evolved magmas must be created or mixed only at shallow depths.

Line 369: any estimate for the minimum basalt flux needed to maintain the mush system? There are a number of numerical modeling studies, by Annen and others more recently. I think they mostly focus on arc systems, but they might indicate the kinds of fluxes needed to maintain a mush above its solidus.

Lines 392-430: I assume that this material is for an Appendix? Otherwise perhaps it should be included much earlier in the article,

Response to Reviewers

Specific questions asked by the three referees are addressed and clarified below, with our responses in grey.

Reviewer #1

1. 80. *This entire Section refers to samples and petrography. Therefore, I would focus the related Fig. 3 only on mineral textures and compositions, avoiding the presentation of different data, such as the barometric estimates. This information can be presented only in the text.*

We have removed the barometric estimates from Fig. 3a. This information is now presented only in the text (line 302). As the reviewer suggests (here and in Comment #6), mineral compositional data are retained on the figure, as these are relevant to the petrographic discussion (e.g. identifying zoning textures).

2. 105. *It is not clear how Mg#_{liq} has been calculated with respect to the iron speciation. Is total Fe expressed as Fe²⁺? There is also confusion with l. 410 stating that melt Fe²⁺/Fe* = 0.85. Is this redox state taken into account only for mineral-melt equilibria or also for the calculation of Mg#_{liq}?*

We made a special effort to be robust in our treatment of Fe speciation. Peterson et al. (2015) measured the Fe²⁺/Fe* ratio of Galapagos glasses. We use this ratio to determine Fe²⁺ and Fe³⁺ concentrations and include only ferrous Fe in our calculation of Mg#_{liq}. No measurements of Fe speciation have been made in minerals and we want to avoid making comparable (risky) assumptions. We therefore calculate Fe²⁺ and Fe³⁺ concentrations by stoichiometry and use the sum of these (Fe*) in our calculations of Mg#_{cpx} and Mg#_{ol}. These are standard redox treatments in Mg# calculations but are not often stated – we want to ensure transparency in our approach.

We have clarified this in the Methods section (lines 423–427), directing readers to the additional details at the appropriate points in the text (lines 106–170 and 143).

3. 129. *Crystal-melt equilibria have been evaluated and presented without explaining before in the text the intra-crystal compositional changes. It is therefore not clear at this stage whether crystal are chemically zoned or not and how their compositions change from core to rim. As a consequence, the statement reported at l. 175 concerning the equilibration of crystals with compositionally distinct melts, appears quite confusing because crystal-melt equilibria has been treated without informing the reader about the crystal zonation. I understand that this makes easiest the methodological approach and I agree with the strategy adopted by the authors. However, I think that it is more logic to present the mineralogical aspects earlier in the manuscript. One possibility is to discuss the mineral chemistry after the Section entitled “Samples and Petrography”. In this new Section you could present most of the compositional changes of minerals reported from l. 180 to l. 220.*

We have added a new section entitled “heterogeneous mineral compositions” on lines 132–202. This introduces the mineral compositional data (including zoning) before assessing crystal-melt equilibria. Rather than adding the new section after “samples and petrography”, we have moved it down slightly and added it after “homogeneous liquids” – in this way, the manuscript progressively introduces evidence of increasing chemical heterogeneity in the seemingly-monotonous magmatic systems.

4. 226-236. *I have greatly appreciated this modeling approach. However, I would interpolate (to some extent) the compositional change of plagioclase with the chemistry of olivine and clinopyroxene. You correctly explain that olivine may suffer for re-equilibration phenomena with compositionally distinct melts. However, I would test whether MELTS simulations may reproduce natural clinopyroxene compositions. I would also report in the text if MELTS modeling indicates clinopyroxene + plagioclase*

saturation at certain P-T-H₂O conditions and if your reconstructed melt compositions are in equilibrium with both plagioclase and clinopyroxene. It is also apparent that most of your discussions and conclusions are based on MELTS simulations. For this reason, I would recommend to provide to the reader your MELTS data as supplementary material, possibly with the temperature-dependent plagioclase-melt partition coefficients calculated with the model of Nielsen et al. (2017). I agree that the incorporation of Ti in plagioclase may be a powerful tool for unraveling cryptic magma dynamics, but I think that your entire approach would appear more robust by combining a little bit different textural and chemical evidences from olivine, clinopyroxene and plagioclase.

We are pleased that the reviewer is supportive of our modelling approach. We have added a new Supplementary Table (Table S7), which contains the liquid and plagioclase compositions calculated by our Rhyolite-MELTS models, as well as D_{Ti} and plagioclase Ti concentrations calculated using the model of Nielsen et al. (2017). We have also produced a new figure comparing our Rhyolite-MELTS models and natural clinopyroxene compositions (Fig. S3). The data in this figure are scattered (our analytical approach wasn't designed for this type of modelling) and there are known problems with Rhyolite-MELTS modelling of clinopyroxene (e.g. Gardner et al. 2014 Contrib. Min. Pet., Burgman & Till 2019 Am. Min.). Consequently, although the model shows a good correlation with our data, we feel that full interrogation of this figure in the main body of the manuscript would result in overinterpretation. Instead, we include it as a supplementary figure to provide readers with additional confidence in our approach. Rhyolite-MELTS only model's major components in olivine and, as the reviewer notes, these undergo significant diffusional re-equilibration, preventing comparison of models (which predict compositions at the time of crystallisation) with natural data (which record the diffusive overprint).

We have added the predicted temperatures of plagioclase and clinopyroxene saturation to the captions of Figures 7, 8 and S3, where the appropriate models are presented (all our models are run at fixed starting H₂O concentrations). We have also noted in the text that our predicted evolved melts are saturated in plagioclase and clinopyroxene (line 254).

5. 348-360. *This is an excellent example of a well-developed scientific discussion.*

We thank the reviewer for this positive comment.

6. *Fig. 3. I suggest to focus the panel (a) only on mineral texture and chemistry. Please identify the plagioclase in panel (b). Please provide An# and Mg#cpx for the numbers reported on the images.*

Done.

7. *Fig. 7. Please add (a), (b) and (c) in the figure caption.*

Done.

8. *Fig. 8. The same as Fig. 7.*

Done.

9. *Fig. 9. Please argument in the caption the compositions of the two melt end-members or, alternatively, report these compositions as supplementary material.*

We have now included these end-member compositions in a new Supplementary Table (Table S8), which is referenced in the figure caption and main manuscript text.

10. *Fig. 10. It would be great to see in this cartoon some indications on P, T, depth (in km) and, perhaps, melt-H₂O contents, if you can easily retrieve these information from previous literature studies.*

We have added bars to Fig. 10 showing relative magma storage depth estimates at Wolf and Fernandina from the literature (separated into geophysical [InSAR] and petrological estimates). There are currently insufficient constraints on temperature or melt H₂O content to adequately illustrate how these vary with depth.

11. *I would not use kbar but rather MPa for the pressure estimates. kbar and Kelvin units are specific to thermodynamic studies calibrating models. Rather, the final target of your work is to understand magma dynamics.*

We have converted kbar to MPa throughout the manuscript (including the Supplementary Information and figures).

Reviewer #2

1. *Line 47: I suggest using “constant range of temperature and composition” rather than “constant”.*

Although we take the reviewer’s point, we find their suggested wording slightly confusing. Instead, we have changed the sentence to “maintaining the system within a narrow temperature and compositional range” (line 48–49).

2. *Line 104: extending ‘the’ database*

This sentence was superfluous and has been removed.

3. *Lines 136-137: I suggest mentioning here that the range in An# is recorded as zonation within individual crystals (based on Fig. 6), other than between distinct crystals.*

This paragraph is discussing the full compositional range of our dataset (both within individual crystals and between distinct crystals). We believe that this is clear now that the data are discussed within the new “Heterogeneous mineral compositions” section (Reviewer #1, Comment #3), where zonation is outlined immediately after the overall trends.

We have noted that “crystals can have anorthite contents covering almost the full range identified in our Wolf samples” on lines 185–186.

4. *Line 309: it is debatable whether the generation of volatile overpressure is sufficient to trigger an eruption, I would suggest avoiding the mention here. Overpressure by second-boiling actually increases the volatile saturation limit of the melt, which leads to the exsolved volatiles partly re-dissolving in the melt and a natural buffer situation is reached. The authors can check the recent work of Townsend et al 2019 GGG (Magma Chamber Growth During Intercaldera Periods), specifically fig 2b for this effect. This paper also shows that exsolution of the MVP corresponds to a sharp decrease in the eruption frequency, similarly to the work of Degruyter et al 2017 GGG (Influence of Exsolved Volatiles on Reheating Silicic Magmas) and Popa et al 2019 JVGR (A connection between magma chamber processes and eruptive styles revealed at Nisyros-Yali volcano). Unzipping by magic recharge is probably the only way upper-crustal reservoirs erupt, including in water-supersaturated rhyolitic systems (e.g. Popa et al., 2019 JVGR).*

We acknowledge the debate around the prevalence of second boiling as an eruption trigger. However, there is abundant petrological (e.g. Stock et al. 2016 Nat Geo, Tramontano et al. 2017 EPSL, Forni et al. 2018) and modelling (e.g. Blake 1984 JGR, Tait et al. 1989 EPSL, Edmonds & Woods 2018 JVGR) evidence to support its occurrence under certain circumstances, including in the reviewer’s recommended paper (Townsend et al 2019). As such, we prefer to retain second boiling as just one possible option in our speculative list of triggering mechanisms for more explosive Galapagos eruptions. This is a very minor point, but we have added an additional supporting reference on line 318 to draw readers attention to the literature on this topic.

5. *Lines 378-380: It is linked to the comment above. There is additional complexity related to the effect of second boiling during magmatic storage and eruptive styles, and it does not necessarily lead to explosive events. Water-supersaturated magmatic reservoirs can sometimes trigger explosive events, but recent work has shown that volatile-supersaturation at storage pressure will more often favor effusivity. Volatile supersaturation creates counter-intuitive feedbacks related to increasing the volatile saturation limit of the melts and increasing the bulk-compressibility of the magmatic reservoir, which allows higher volumes of recharge melt to be accommodated before a state of critical overpressure is reached. This in turn allows*

for efficient reheating and melt viscosity drop, which favors outgassing. Moreover, once the eruption is triggered nucleated volatiles are already present at the base of the conduit. Previously re-dissolved MVPs (under disequilibrium conditions caused by the overpressure of mafic recharge being accommodated in the upper-crustal reservoir) re-nucleate on-masse creating a foamy melt at the onset of conduit ascent. The presence of the foam combines with the reheating and viscosity reduction to favor conduit outgassing, which can neutralize the explosive potential of the magma. For more details, the authors can check the papers mentioned above (Degruyter et al., 2017; Popa et al., 2019, Townsend et al., 2019).

We thank the reviewer for this information. It does not seem that they are requesting any changes to the manuscript, but we have changed the wording to clarify that second boiling “might” generate explosive silicic eruptions (line 392). At this stage, we do not have enough information to differentiate between the effusive and explosive scenarios outlined in their comment.

6. Figure 3 caption, line 637: I think it is (c,d,e) rather than (d,e,f)

The reviewer is right – this has been corrected in the manuscript.

7. Figure 7 and 8 captions: are a bit confusing. Plagioclase TiO₂ vs An# appear in plots (c) on both figures, while they are announced at the beginning of the captions. Maybe TiO₂ vs An# should be plots (a) instead? Also, the figure captions do not distinguish between plots a,b,c.

Labels have been re-ordered so that plagioclase TiO₂ vs An# is plot (a) in both figures. (a), (b) and (c) have been added to the figure captions.

Reviewer #3

1. The one thing I was left curious about is what kinds of magma supply rates are needed (thermally) to sustain the mush system above its solidus, and (compositionally) to flush the system with enough basalt to keep the felsic:mafic mixing ratios low. The authors need not conduct their own modeling, but I'm wondering if any of studies such as Annen (2008), Schopa and Annen (2013), Karakas and Dufek (2015), or Jackson et al. (2018), etc., lead to any insights about plausible Galapagos melt supply rates. I know these numerical studies are mostly focused on the generation and maintenance of silicic systems, but perhaps they are still useful in the present case. I think the discussion would be aided by some mentioning of supply rates.

The preservation of a thermally stable super-solidus crustal mush zone is not controversial and has recently become a well-accepted model in volcanology (e.g. Cashman et al. 2017 Science). We have carefully reviewed the literature to answer this question but, as the reviewer suggests, we found that most of the modelling studies to date have focused on the generation of long-lived mush zones in silicic continental settings. Nevertheless, recent reviews of the available thermal models (Bachmann & Huber 2016 Am Min, Cooper 2019 Phil Trans Royal Soc) have identified that long-lived mush zones can exist with melt fluxes $>10^{-4} \text{ km}^3 \text{ yr}^{-1}$, even in the upper crust.

The long-term eruptive fluxes at Wolf and Fernandina are on the order $0.4\text{--}1 \cdot 10^{-3}$ and $4.4 \cdot 10^{-3}$, respectively (Geist et al. 2005 J Pet, Kurz et al. 2014 AGU Monograph), suggesting that the magma supply rates are sufficient to preserve a super-solidus lower crustal mush (thus addressing the reviewers comment).

Although eruptive fluxes have not yet been quantified for compositionally heterogeneous volcanoes in the eastern Galapagos Archipelago, we note that Harpp et al. (2014 AGU Monograph) estimate them to be several orders of magnitude lower than in the western archipelago (e.g. at Wolf and Fernandina). In this case, the crustal magma flux is unlikely sufficient to preserve super-solidus conditions beneath eastern Galapagos volcanoes. This provides additional support for our conclusion that the compositional diversity of erupted products is dictated by the crustal melt flux.

We have added this information on lines 382–385,

2. *The only other issue, and it's a minor one as I know this is not a key aspect of the study, is that the pressures of magma storage are probably not well constrained by the methods employed. The key assumption is that any particular whole rock trend represents an isobaric one. And MELTS or other models can provide such a P. But that inferred P may have little connection to the conditions of crystallization (it need not even be an average pressure). And yet the authors have clinopyroxene, so it seems that it should be possible to estimate at least rough crystallization depths (P estimates are always rough), which based on other studies of oceanic islands systems, may be much more polybaric than indicated here (see Klugel and Klein 2005 at Madeira; Putirka 1996 at Hawaii).*

I am not sure that Ti-in-plag is sensitive enough to P to provide a useful barometer. I may be wrong, but P is very difficult to estimate using any condensed-phase equilibria, and I think estimates are almost as good as useless in the absence of plots of P(measured) vs. P(calculated) that indicate error for the method in play (where "measured" is for experimental systems, where P is known). I don't see those kinds of tests in Nielsen et al. (2017), and when I look at Nielsen et al.'s Table 6, I don't even see a P coefficient for D(Ti), which would mean that even the D is perhaps not all that sensitive to P; but perhaps I'm just missing something.

The only other possible "barometer" at work is implicit in the MELTS simulation of ilmenite saturation. But MELTS is only calculating a very small apparent change in phase appearance (e.g., based on the shift in the peak in the Ti v An curve in Figs 7-8), and even that peak is probably not a useful barometer, if one were brave enough to plot P(measured) vs. P(calculated).

The MELTS curves collapse onto one another at the low K₂O/TiO₂ ratios that characterize Wolf, and they miss the higher K₂O/TiO₂ parts of the Fernandina array, which the authors convincingly attribute to magma mixing. But this mixing also means that if we plotted all of the major oxides we would likely find that the MELTS-derived model liquids do not match the observed whole rock trends, which they must, since the whole rocks are mostly liquid.

The problem can be solved by using observed whole rock compositions - and the extension of such trends - as a model for plausible liquids, as I note in the comments below. If we are not dealing with antecrysts, then the liquids relevant to the mineral compositions must either fall on the whole rock trends, or on their extensions to lower or higher Mg#. By using mixing, or fractional crystallization or AFC, etc. - whatever it takes to describe and extrapolate the trends - the authors can use such trends to see if any composition along such can explain the observed mineral compositions. If the authors are interested, they can see the method in play in Scruggs and Putirka (2018). In any case, here, it looks like mixing is a dominant process and, this, not MELTS, would be the most appropriate way to estimate liquid compositions for observed mineral phases. Precisely how the curve is created, though, does not matter (except in the extrapolated regions), so long as the whole rock trend is reproduced. The method fails for antecrysts, but antecrysts can be identified by using multiple tests of equilibrium. Taking cpx as an example, an antecryst might satisfy Fe-Mg exchange equilibrium for a putative liquid, but then fail to match predicted equilibrium values for DiHd, EnFs, etc. This method is not fool-proof. We can never be sure that the crystals that pass such tests are phenocrysts. They could be antecrysts from an older event, if the liquids from an earlier episode formed along the same computed liquid trend. But in such a case the information is still useful, as to produce the same crystals, the system would have to crystallize at the same P-T conditions.

Such an approach may well yield a narrow range of pressures for the Wolf and Fernandina systems. But if I were to be money, it would be that the crystals reveal a wide, rather than a narrow range of pressures. But in any case, I don't think their pressure estimates are the main focus of this work. They have a very good case for heterogeneous liquids, and I think they support the mush-filtering concept; if the authors agree that the above arguments have some merit, they can make some tests to consider a more polybaric system. But I don't know that such an analysis should be required, though in the absence of tests, they can just say something like "If we take the Ti partitioning data and MELTS modeling at face value, then it would appear that pressures of crystallization are in the range 2.3-3 kbar..."

We thank the reviewer for this comprehensive and supportive comment. However, it seems like our writing was misleading as we do not rely on either plagioclase Ti partitioning or MELTS ilmenite

saturation for barometric estimates. We have undertaken a comprehensive study of magma storage depths/pressures at Wolf volcano using petrological barometry and geophysics. This was the focus of a separate contribution (Stock et al. 2018 G³). At Fernandina, we have found insufficient equilibrium clinopyroxene-melt pairs to replicate this barometry, but geophysical constraints support a similar sub-volcanic architecture (Bagnardi et al. 2013 G³) and geochemical analyses suggest slightly higher magma storage pressures (Geist et al. 2006 G³). Given these detailed existing studies, it is unnecessary to apply the Scruggs and Putirka method (further discussion of this method is in response to Comment #14, below).

Rather than using our models to infer magma storage pressures, we have used these previous constraints to guide our modelling approach. As the reviewer predicts, our models are largely insensitive to pressure. However, in testing the impact of this parameter, we found that the best results are produced at the previously identified crystallisation pressures. We believe that this validates our model inputs (i.e. the outputs are consistent with previously published, more accurate pressure constraints) but did not intend to suggest they provide any kind of barometric estimate.

We have attempted to clarify this by stating that our choice of input pressures was guided by independent constraints and that the models are relatively insensitive to pressure on lines 240–242. As the reviewer suggested, we also reflected this insensitivity by adding the phrase “taken at face value” when comparing model pressures with independent constraints (lines 242 – 243).

3. *Line 38: “...distinct batches of magma interact...”; this model will not be familiar to most readers. Expand by noting that the magmas “...interact with gabbroic wall rock”. Also, in reading Bedard (1993) and also Geist et al (2005), I think that the Geist paper, and/or Sinton and Detrick (1992), would be a better citation. To my reading, those papers make a clearer case for how eruptive monotony might be imposed on a system that maintains a crustal mush above its solidus. I don’t think that monotony is an obvious result of assimilation or reaction with gabbros; but maintenance of a mush above its solidus seems plausible, and probably connects to some critical, and perhaps also near-constant, melt supply rate...?*

We removed the Bedard (1993) reference and instead cite this model using Geist et al. (2014; which builds on the ideas presented in Geist et al. 2005). We have expanded the sentence on lines 39–40 to say “...interact with surrounding gabbroic material”. As the reviewer points out, monotony would typically require the gabbro to be above the solidus. For this reason, we avoid adding the term “wall rock” (which, to us, implies solid material) and added the clarifying term “super-solidus mush” on line 42.

4. *Lines 39-40: Do you mean, here, the putative density “filter”, where buoyancy controls whether (or disallows) primitive magmas move to the surface? As I am sure the authors well know, Stolper and Walker (1980) showed that MORB appeared to populate a density minimum, and they argued that that density corresponded to that imposed by the density of rocks near the Moho. Glazner (1995) made an analogous argument for continental magmatic rocks, but arguing that the middle crust was the key density filter. Using newer models for magma density, I don’t see a density minimum (see Figure 3 in Putirka, 2017; Elements). So I would argue that there is a filter, but only against ‘primitiveness’, not against diversity. I think this is what Bedard is arguing for also – that gabbro-melt reactions provide a filter against “primary” magmas, but a filter that does not preclude the eruption of diverse, non-primitive magmas.*

Yes, on lines 40–41 we are essentially describing variants on the Stolper and Walker (1980) model and have added this citation. We removed the Bedard reference in response to Comment #3, above.

5. *Line 40: Is the argument here that some volcanoes might erupt homogeneous magmas due to differentiating at similar P-T conditions? The argument also implies that they differentiate to the same extent. So it’s the critical density argument of Stolper and Walker (1980), right?*

Yes, we have added this citation on line 41.

6. *Lines 47-48: Wouldn’t constant melt input – a constant melt supply rate, after some initial start-up, lead to a near “exact balance” of heat input and output?*

A constant melt supply rate wouldn’t lead to a near exact thermal balance if the heat input was greater or less than the heat loss by advection and/or eruption. For example, even if the melt flux was constant, the

system would cool and evolve if the heat supplied was lower than that loss by advection. We have attempted to clarify this by stating explicitly that “monotonous activity requires an exact thermal balance...” on line 49.

7. *Lines 97-98: “with clinopyroxene lining many of the void walls”. Does this imply that at least some cpx grains precipitated from a vapor phase? Or only that the vapor-containing voids formed from a cpx matrix? It seems like the former is the case. Out of pure curiosity, having nothing to do with the issues here, I would be curious to know if these cpx are different or distinct in composition in any particular way.*

We avoided analysing these pyroxenes as we wanted to concentrate our investigation on melt processes (as opposed to the vapour phase). However, we do imagine that this clinopyroxene is associated with a hydrous fluid or a very late-stage melt. The reviewer can find more information about these features in Sisson et al. (1996, CMP).

8. *Line 104: insert “the” before “Harpp”.*

This sentence was superfluous and has been removed (same response as Reviewer #2, Comment #2).

9. *Line 109: and accumulated plag also affects whole rock MgO, and accumulated olivine affects Mg#, right?*

This is correct and we have clarified the statement on lines 109. Accumulated plagioclase would decrease the whole-rock MgO and FeO but this effect is very small (<1% change in Mg# for 10% accumulated feldspar). Accumulated olivine significantly increases bulk-rock Mg#. In either case, accumulated crystals would increase the Mg# interquartile range of subaerial lavas, making the real erupted liquids even more homogeneous than these data suggest.

10. *Lines 114-115: Perhaps a callout to Fig. 3 here, and for comparison, perhaps a light, dashed line in the histogram in Fig. 3 could indicate the range in Mg# from another Galapagos volcano, that is not monotonous, since monotony is to some extent, relative?*

Done (the reviewer meant Fig. 2).

11. *Line 119: cite standard deviation of Mg#; also cite, avg. \pm std dev for tephra.*

Done.

12. *Line 121: cite avg \pm std dev. For scoria glass.*

Done.

13. *Line 158: do you mean “heterogeneity of sub-volcanic melts”?*

Yes, this has been updated in the text (now on line 204).

14. *Lines 158-160. Why not go the other direction, as that would get more directly at your stated goal? See Scuggs and Putirka (2018; Am Min). The method is simple, and likely much more accurate. The basic idea is that 1) you use observed whole rock and glass compositions, and a mixing line (or a fractionation curve) to interpolate or extrapolate from these. (Yes, whole rocks are not liquid, but they mix with liquids; most lavas have <20% crystals so they are 80% liquid; so, liquids must lie along any mixing or differentiation trend that contains whole rock compositions). 2) Now, find out of any liquids along such a trend can satisfy tests of equilibrium for your observed mineral compositions. These are the liquids that existed in the sub-volcanic system.*

We see the value of this approach but do not believe that it would work in this case. From the reviewer’s comment and the Scuggs and Putirka (2018) paper, our understanding is that the method defines a mixing/fractionation line from all available erupted liquids, and then tests for equilibrium between erupted crystals and the liquids along this line. This works well at volcanoes (such as Chaos Crags) that have erupted compositionally heterogeneous liquids and where erupted crystals have grown from liquids within this compositional range, but not for monotonous volcanoes where crystals grew from liquids that are significantly different to those that have erupted at the surface.

In the case of Wolf and Fernandina, the erupted liquids are near-homogenous and mafic (Fig. 2). Consequently, they only define a liquid line of descent over a limited, primitive compositional range. Our analyses show that crystals grew from sub-volcanic liquids that are significantly more evolved than any material erupted at the surface. Hence, to apply the Scuggs and Putirka (2018) method, we would need to extrapolate significantly from the whole-rock and glass data in order to define a liquid line of descent down to much more evolved compositions. It might be possible to define fractionation/mixing lines using data from other heterogeneous Galapagos volcanoes, but the magmatic systems beneath these volcanoes are often poorly understood (e.g. Alcedo) and there is significant variability in primary melt compositions across the archipelago (e.g. Harpp & Geist 2018 *Frontiers*); we are therefore reluctant to rely on data from totally different systems to interpret Wolf and Fernandina, as there may be significant variability.

In the present study, we believe that our approach of defining the liquid line of descent using thermodynamic models (i.e. Rhyolite-MELTS) is more robust than Scuggs and Putirka (2018), particularly as crystals record the appearance of mineral phases (e.g. ilmenite) which are outside the compositional range of erupted liquids. We acknowledge that Rhyolite-MELTS models are not perfect, but the good correlation between the modelled and measured mineral (Figs. 7, 8, S3) and liquid (Figs. 2, S1, S2) compositions suggest that the models are reliable in this case. Similarly, in response to Comment #2 (above), we note that while mixing is detectable in a minority of Fernandina eruptions (Fig. 9), all mineral and most liquid data (including the 2015 and 1968 eruptions) can be modelled by pure fractional crystallisation. This is the premise of our “basalt flushing” argument – while mixing does occur, the evolved end member is in such low abundance that the mass balance makes it undetectable and magmas follow a normal fractionation trend (lines 363–365).

Our reasoning is verified by the reviewer in Comment #20 and demonstrated in response to Comment #22, below. We have noted the advantage of our method over that of Scuggs and Putirka (2018) on lines 443–445.

15. *Lines 161-164: If these calculated average mineral compositions are meant to be compared to the averaged noted in the prior paragraph, most of this text and the prior paragraph can be deleted - substituted by and more effectively presented in a table.*

We would like readers to compare the full distribution of the measured mineral compositions with these calculated values. The data are most effectively presented in Fig. 4 but we believe that retaining a discussion of the data distribution is necessary in the text. To clarify this, we have noted that the calculated compositions are compared with the “compositional distribution” of our data and have added an extra reference to Fig. 4 on lines 206–207.

16. *Lines 203-205: some of these patterns are intriguing, such as a “small decrease in An” accompanied by increases in K and Mg. I would think that An and Mg would be positively correlated, and that increases towards the rim would indicate recharge. But recharge should cause K to drop. Could the inverse of the anticipated trends be due to an increase in growth rate, where K and Mg just can’t get out of the way fast enough as the crystal/liquid interface moves rapidly towards the liquid?*

Although we think it is most likely that the anticorrelation between An# and MgO is due to the high Mg partition coefficient in more albitic plagioclase (see response to Comment #24, below), we cannot discount that a small number of analyses close to crystal rims have elevated MgO and K₂O concentrations due to the formation of compositional boundary layers during rapid growth. We have noted this on lines 289–290, with a supporting citation (Honour et al 2019 *Nat Comms*).

17. *Line 214 “...in which An# correlates negatively with K₂O.” That is expected, right, for “normal zoning”?*

Yes, we have removed this superfluous statement.

18. *Line 222: Perhaps cite a possible T here? I've done some calculations recently using Grove et al. (1984) and the rates are very slow; it seems that once plag has crystallized, it won't really re-equilibrate except at the sub-micron scale.*

We have removed this sentence in response to Comment #19, below. The Grove et al. (1984) paper is about CaAl-NaSi interdiffusion, which we agree is very slow even at magmatic temperatures (lines 226–227).

19. *Lines 224-226. I agree, but I think you can delete all of lines 222-226 and then at the end cite Grove et al. (1984) and the new Cherniak and Watson paper at the end of the sentence.*

We think that the reviewer might have been referring to the first two sentences in this paragraph (i.e. including the sentence queried in Comment #18, above). We agree that these were superfluous, and they have been removed. We think that it is important to state that An# and TiO₂ are slow diffusing in plagioclase, as this is fundamental to our interpretation. Consequently, we have retained this sentence and given the reviewer's recommended citations (lines 226–228).

20. *Lines 231-232: This is not bad, but the advantage of the method that I suggest (see comments for lines 158-160) is that you don't have to worry about whether your trends are produced by fractional crystallization, or mixing or assimilation, or affected by recharge, etc. Here, by relying on MELTS, your calculated liquids are only accurate to the extent that the range of liquids that produced your observed plagioclase crystals also formed by pure fractional crystallization. For moderately compatible elements like Mg or Fe, mixing and FC are both pretty linear. But for incompatible elements there is a lot of curvature to the FC trend, and so the calculated liquids may lie quite far from the actual rocks, if compositional trends are affected by magma mixing. If you instead start with observed whole rock trends (which liquids must follow) and then ask what compositions along these trends – if any – can explain my observed mineral compositions? Then you eliminate any assumptions about the processes by which the whole rocks/liquids were generated. This method only fails if the liquids that generated your observed minerals are not on the whole rock trend. But that can only be the case for those crystals that are not part of the evolutionary suite you are examining (e.g., the method fails if you have xenocrysts, rather than autocrysts). You won't discover that through MELTS, but you can often detect that error if mineral-melt equilibrium tests do not provide consistent results between clinopyroxene and plagioclase. (Mineral-scale isotopes will also reveal the problem, but that's an expensive and time-intensive approach).*

We are pleased that the reviewer generally agrees with our modelling. In this study, however, we are confident that our approach is preferable to the proposed method, as detailed in response to Comment #14 (above).

Importantly, the reviewer verifies our reasoning in this comment: many of the crystals that we are examining do not fall on the whole-rock (or glass) evolutionary trend, as they formed from melts that are significantly more evolved than any magmas erupted at the surface. We don't identify this from Rhyolite-MELTS (or isotopes) but employ a similar method to Scuggs and Putirka 2018, calculating the compositions of crystals that would be in equilibrium with the carrier liquid (lines 430–435) and showing that many of our analyses extend to more evolved compositions (i.e. they cannot have formed from that liquid and are not autocrysts).

Our crystals could be called “antecrysts” or “xenocrysts”, but we have deliberately avoided these terms, as they make genetic assumptions. Instead, we prefer the term “macrocrysts”. The presence of glomerocrystic aggregates (i.e. disaggregated mush) provides further evidence that some of the crystal cargo is inherited. However, Rhyolite-MELTS models support that almost all magmas at Wolf and Fernandina are related by fractional crystallisation, even if they aren't represented in the erupted record (see response to Comment #14). We have now added this reasoning on lines 447–453.

21. *Lines 238-239: you have pyroxenes, so why use MELTS to guess at pressure when you can calculate it? Then run Rhyolite MELTS at the calculated pressures for a better estimate of the plagioclase compositions?*

We didn't use MELTS to estimate pressure, we tested the pressure sensitivity of the model but were guided by previous clinopyroxene-melt barometry (see response to Comment #2, above).

22. *Lines 241-242: Again, you are better off using Ti partition coefficients for plag and then figuring out where along your whole rock trends that you can find a liquid that has the right Ti content to explain the low Ti you see in certain plag. GERM cites several studies that place the Kd for Ti between plag and melt at about 0.04, and no lower. So if this is the lowest reasonable Kd, then what Ti is needed in the liquid to explain your Ti-in-plag values? And do any of your whole rock trends either contain, or extrapolate to such values? If they do, then you have a possible liquid from which these plag crystallized from. If the Ca#(liq) also works out to give you the right An content of the plag, then you're good - you know the liquid rather precisely.*

See responses to Comments #14 and #20, above.

This approach won't work in the present case because plagioclase Ti concentrations record an inflection at *ilmenite-in* but all the erupted liquids from Wolf and Fernandina record ilmenite-undersaturated melts (i.e. Ti and Mg# are anticorrelated in all erupted whole-rocks and glasses; Fig. 2). Ilmenite is present as inclusions in some evolved pyroxenes, verifying the plagioclase data.

As a back of the envelope demonstration: the TiO₂ concentration in our lowest An# plagioclase crystal is ~0.1 (Fig. 7). With a D_{Ti} of 0.04, a liquid TiO₂ concentration of ~2.5 wt% would be required to produce the this plagioclase composition. Although there are erupted Wolf liquids containing ~2.5 wt% TiO₂, they are all very primitive, record ilmenite-undersaturated melts and could not be in equilibrium with low-An# plagioclase. Instead, this crystal must have grown from an evolved melt that is not represented in the erupted whole-rock record. This precludes the Scroggs and Putirka (2018) method and demonstrates why our MELTS-based method is preferable in this case.

23. *Lines 253-256: I think the very low An contents seen at Fernandina (An30s) could very well be co-saturated with quartz. We've done modeling at Chaos Crags where we have An contents that reach down to about An33 or so, and we've found that it's almost impossible to create these in the absence of melts that are about 70% SiO2 or higher. We discovered this by the kind of reverse modeling I've noted above (use the crystal and whole rock trends to discover the equilibrium liquid) and the inferences have lately been confirmed by analyses of melt inclusions and matrix glass, which have the predicted compositions from our low-An data, and are quartz saturated. Of course, the analogy is imperfect, as we used Lassen-derived whole rock trends to make our predictions of the evolved (and un-erupted) liquid compositions (the recent eruptions at Lassen do not involve rhyolitic whole rocks; we infer that rhyolitic magmas exist due to the presence of the low-An plag; this is new work but touched on in Scroggs and Putirka).*

If the low-An# plagioclase was co-saturated with quartz that would agree with our petrological observations and it's possible that the low temperature of *quartz-in* reflects an imperfection in Rhyolite-MELTS. However, in Galapagos, we note that quartz is absent in samples from Alcedo volcano which also have An# = 19–30 (Geist et al. 1995 J Pet) but is present in Rabida crustal xenoliths which have An# ~15 (McBirney & Williams 1969 GSA Memoir). Although these analogies are still not perfect for Wolf and Fernandina (see response to Comment #14, above), they are likely more comparable than Chaos Crags, both in terms of the magma system architecture and parental melt composition. In the absence of direct evidence (e.g. inclusions), we are therefore reluctant to use the Chaos Crags example to interpret that our lowest An# feldspars are co-saturated with quartz. Instead, we prefer to err on the side of caution, suggesting that our plagioclase analyses do not pick up the full compositional diversity of sub-volcanic melts.

24. *Line 272: So Mg partitioning into Plag is inversely proportion to An content? Wow. I would have guessed the opposite. I don't see a clear statement of such in the Nielsen et al paper. Could you give partition coefficients for a Ca- and Na-rich case so we can see what sorts of magnitudes are active here.*

Experiments show a weak anticorrelation between D_{Mg} and An#. This was identified by Bindeman (1998 GCA) and used in the study of Humphreys (2009). Nielsen et al. (2017) re-parameterised the model, carefully filtering the available experimental data for quality (Nielsen et al. don't seem to explicitly discuss the relationship between D_{Mg} and An# but parameters defining the relationship are given in their Table 6). D_{Mg} also depends on temperature. The partition coefficients in our model only vary between 0.02 in the highest temperature, most anorthitic feldspars and 0.05 in the lowest temperature, most albitic crystals (which are within the range of the Nielsen et al. D_{Mg} calibration). We have added this to the text (line 280), along with our calculated values of D_{Ti} (line 236) for consistency.

25. *Line 276: "...melts containing 1-2 wt. % MgO". OK. But if there is magma mixing or wall rock assimilation, you might get a different answer. The MELTS model is only giving you only the liquid trajectory for a highly idealized fractional crystallization case.*

See our response to Comment #14, above. As fractional crystallisation models do a good job of replicating measured mineral (Figs. 7, 8, S3) and liquid (Figs. 2, S1, S2) compositions from the 2015 and 1968 eruptions, we see no evidence for wall rock assimilation. This is consistent with a study by Geist et al. (1995 J Pet), which showed that Galapagos volcanoes evolve primarily through fractional crystallisation with little or no assimilation. We now state this explicitly on lines 451–453. As outlined in response to Comment #14, we use Fig. 9 to identify that only a small number of Fernandina eruptions have undergone substantial mixing. The 2015 and 1968 magmas have undergone some mixing with evolved magmas (as evidenced by the evolved minerals, i.e. antecrysts) but Fig. 9 shows that the proportion of this evolved component is too low to have had a detectable impact on the liquid line of descent defined by erupted magmas. This is detailed on lines 340–354.

26. *Line 295: how do you get these pressures (2.5-3 kbar)? That's pretty high precision. You think you have the depths of equilibration to within a 0.5 kbar range? Perhaps this is from an average, where you are using a standard error? And of course, the method matters. Not all models have the same precision.*

The depth estimates are from the clinopyroxene-melt barometry of Stock et al. (2018 G^3 ; see response to Comment #2, above). Although the individual values have a standard error of ± 140 MPa, they overlap within uncertainty with a lower crustal magma storage region that has been accurately located using integrated petrological and geophysical techniques (Stock et al. 2018). We have clarified this on lines 302–304.

27. *Line 309: This is a minor point, given modern usage, but while I think I've used the term before, I am going to avoid using "second boiling", given what I have since discovered. This term originated with theoretical studies of phase relationships in 2-component systems by Roozeboom (1901), which I think includes some experimental verification in the system KNO_3-H_2O . The case was shown that the triple points of two substances can connect to form a curve that in T-P space exhibits a maximum pressure. Upon heating at constant pressure, it would then be possible to intersect the liquid/vapor (boiling) curve twice, hence the term second boiling. But the processes we are discussing do not involve the connection of two triple points and the double intersection of a liquid/vapor curve.*

I say that this is a minor point because the original phenomenon is so rarely discussed in any context, and the original usage of "second boiling" now so heavily buried in the literature, that it probably doesn't matter at this point how the phrase is used. We are perhaps free to be use it any way we like.

We have carefully considered whether to remove the term "second boiling". However, we believe that it is now synonymous crystallisation-induced volatile saturation in volcanology (and economic geology) and avoiding it might cause ambiguity, inhibiting integration of our findings with the existing literature

(which is particularly important, in response to Reviewer #2 Comments 4 and 5). Instead, we have chosen to retain the term and define it at the first usage (line 317).

- 28.** *Lines 350-360: this all makes perfect sense. I just am not sure whether the evolved magmas must be created or mixed only at shallow depths.*

Some low An# plagioclase cores have fully concentric high An# mantles and are in glomerocrystic aggregates with high-P clinopyroxene (based on clinopyroxene-melt barometry; Fig. 2). The glomerocrysts are interpreted as fragments of disaggregated mush and the fully concentric mantles suggest that mixing occurred before crystals were incorporated into a cumulate pile. The simplest explanation is that the cumulates derive from the lower crust (based on the barometry), and so at least some of the mixing likely occurred at depth. We have noted this on lines 338–339.

- 29.** *Line 369: any estimate for the minimum basalt flux needed to maintain the mush system? There are a number of numerical modeling studies, by Annen and others more recently. I think they mostly focus on arc systems, but they might indicate the kinds of fluxes needed to maintain a mush above its solidus.*

See response to Comment #1, above.

- 30.** *Lines 392-430: I assume that this material is for an Appendix? Otherwise perhaps it should be included much earlier in the article.*

The Methods are at the end of the article, as per the Nature Communications style.

REVIEWERS' COMMENTS:

Reviewer #1 (Remarks to the Author):

Dear Editor,

I have found the revised manuscript substantially improved and my early concerns well addressed in the rebuttal letter. For this motivation, I'm pleased to suggest publication of this noteworthy work in its current form.

Sincerely,
Silvio Mollo

Reviewer #2 (Remarks to the Author):

There is nothing more that I would like to add. The paper is in a good shape and, from my point of view, ready for publication.

Reviewer #3 (Remarks to the Author):

The authors did an excellent job of responding to comments. I accept their rebuttals and just have one remaining question, relative to their Figure 9.

In that figure, curves represent liquid paths during fractional crystallization. A subset of whole rocks follow the flatter, low-K/Ti portions of these trends. The observed whole rocks (referred to as "erupted liquids" in an earlier panel) reflect mixing between primitive and evolved compositions. The mixing envelope is calculated assuming a single evolved mixing end-member and a range of low-K/Ti, primitive end-members.

What is special about compositions near the green dots (2015 whole rocks), such that the high K/Ti end member chooses to mostly mix with that composition, and not others? In the lower panel, for example, why is that high-K/Ti (= ca. 1.9) evolved magmas never see magmas with Mg# between 30 and 50?

If the fractional crystallization curves are really telling us about liquid evolution, then shouldn't the evolved magmas interact with liquids all along these lines?

I'm not sure whether this is an easy or difficult question to answer, but in either case, I think the paper is ready for publication.

Response to Reviewers

The question asked by Reviewer #3 in the second round of reviews is addressed below, with our response in grey.

Reviewer #3

1. *The authors did an excellent job of responding to comments. I accept their rebuttals and just have one remaining question, relative to their Figure 9.*

In that figure, curves represent liquid paths during fractional crystallization. A subset of whole rocks follow the flatter, low-K/Ti portions of these trends. The observed whole rocks (referred to as “erupted liquids” in an earlier panel) reflect mixing between primitive and evolved compositions. The mixing envelope is calculated assuming a single evolved mixing end-member and a range of low-K/Ti, primitive end-members.

What is special about compositions near the green dots (2015 whole rocks), such that the high K/Ti end member chooses to mostly mix with that composition, and not others? In the lower panel, for example, why is that high-K/Ti (= ca. 1.9) evolved magmas never see magmas with Mg# between 30 and 50?

If the fractional crystallization curves are really telling us about liquid evolution, then shouldn't the evolved magmas interact with liquids all along these lines?

I'm not sure whether this is an easy or difficult question to answer, but in either case, I think the paper is ready for publication.

In panel a), all the erupted liquids from Wolf sit on the low K/Ti portion of the fractional crystallisation curves. Perhaps a small number of datapoints at lower Mg#_{liq} show slightly elevated K/Ti ratios and record mixing with the evolved endmember, but these are taken from the literature and without being able to inspect the samples we are reluctant to overinterpret the data. In panel b), several Fernandina whole-rock analyses from the literature have elevated K/Ti ratios and record mixing between primitive and evolved endmembers. As the reviewer notes, these have a restricted range of Mg#_{liq} (~40–55) and do not record mixing between the evolved endmember and primitive endmembers extending all along the fractional crystallisation curves.

All the Fernandina erupted liquids that show evidence of mixing derive from a small number of submarine vents located on the southwest flank of the volcano. The close spatial relationship of these lavas suggests that they were fed by liquids ascending from the same part of the sub-volcanic plumbing system and bathymetric data suggest that they were erupted at a similar time (see Geist et al. 2006, G³). Hence, the primitive endmembers ascending to feed these eruptions were likely compositionally analogous, with a similar Mg#_{liq}. We can only speculate why primitive liquids from this part of the magmatic system mixed with more of the evolved end member (increasing their K/Ti ratio). As they are distal from the centre of the volcano, one possibility is that the melts ascended through a colder part of the sub-volcanic system where larger volumes of low temperature evolved magma were able to accumulate.

We have clarified this on lines 355–358 of the revised manuscript.